# ESRP1-regulated isoform switching of LRRFIP2 determines metastasis of gastric cancer

Jihee Lee[1,2,11], Kyoungwha Pang[1,11], Junil Kim[3], Eunji Hong[1,4], Jeeyun Lee [5], Hee Jin Cho[6,7], Jinah Park [1], Minjung Son[1,4], Sihyun Park[1], Minjung Lee[8], Akira Ooshima [1], Kyung-Soon Park [2], Han-Kwang Yang[9,10], Kyung-Min Yang[8] & Seong-Jin Kim [1,8] ✉

Although accumulating evidence indicates that alternative splicing is aberrantly altered in many cancers, the functional mechanism remains to be elucidated. Here, we show that epithelial and mesenchymal isoform switches of leucine-rich repeat Fli-I-interacting protein 2 (LRRFIP2) regulated by epithelial splicing regulatory protein 1 (ESRP1) correlate with metastatic potential of gastric cancer cells. We found that expression of the splicing variants of *LRRFIP2* was closely correlated with that of *ESRP1*. Surprisingly, ectopic expression of the mesenchymal isoform of LRRFIP2 (variant 3) dramatically increased liver metastasis of gastric cancer cells, whereas deletion of exon 7 of LRRFIP2 by the CRISPR/Cas9 system caused an isoform switch, leading to marked suppression of liver metastasis. Mechanistically, the epithelial LRRFIP2 isoform (variant 2) inhibited the oncogenic function of coactivator-associated arginine methyltransferase 1 (CARM1) through interaction. Taken together, our data reveals a mechanism of LRRFIP2 isoform switches in gastric cancer with important implication for cancer metastasis.

Alternative splicing is a prevalent mechanism in complex organisms by which multiple protein isoforms can be produced from a single gene, generating protein diversity. It creates multiple mRNA variants from a single gene through the selection and utilization of alternative splice sites in the pre-mRNA via different splicing events[1]. Alternative splicing events are tightly regulated by numerous factors, and their dysregulation has been observed in a number of diseases, including various types of cancers.

Epithelial splicing regulatory protein (ESRP1) is an epithelial cell-specific RNA-binding protein that regulates the alternative splicing of multiple genes involved in epithelial–mesenchymal transition (EMT), which has a critical role in metastasis by reducing tumor motility and invasiveness[2–4]. ESRP1 was identified as an essential regulator of fibroblast growth factor receptor 2 (*FGFR2*) splicing, causing a switch in endogenous FGFR2 splicing from the mesenchymal to the epithelial isoform[2]. Additionally, CD44, a cell surface protein, is another prominent target of ESRP1. It has been reported that isoform switching from the variable exon-containing CD44 variant isoforms (CD44v) to the variable exon-absent CD44 standard isoform (CD44s) regulated by ESRP1 is functionally essential for cells to undergo EMT in breast

[1]GILO Institute, GILO Foundation, Seoul 06668, Korea. [2]Department of Biomedical Science, College of Life Science, CHA University, Seongnam, Gyeonggi-do 13488, Korea. [3]School of Systems Biomedical Science, Soongsil University, Seoul 06978, Korea. [4]Department of Biomedical Science, College of Life Science, Sungkyunkwan University, Suwon, Gyeonggi-do 16419, Korea. [5]Division of Hematology-Oncology, Department of Medicine, Samsung Medical Center Sungkyunkwan University School of Medicine, Seoul 06351, Korea. [6]Department of Biomedical Convergence Science and Technology, Kyungpook National University, Daegu 41566, Korea. [7]Innovative Therapeutic Research Center, Precision Medicine Research Institute, Samsung Medical Center, Seoul 06531, Republic of Korea. [8]Medpacto Inc., Seoul 06668, Korea. [9]Department of Surgery, Seoul National University Hospital, Seoul 03080, Korea. [10]Cancer Research Institute, Seoul National University, Seoul 03080, Korea. [11]These authors contributed equally: Jihee Lee, Kyoungwha Pang. ✉e-mail: jasonsjkim@gilo.or.kr

cancer[5,6]. ESRP1 also regulates splicing of p120-catenin (CTNND1) and hMena (ENAH) in the same manner during EMT in several types of cancers[7]. Although accumulating evidence indicates the roles and clinical significance of ESRP1 and its alternative splicing targets in tumor progression and metastasis, ESRP1-regulated alternative splicing in gastric cancer has not yet been thoroughly studied.

Gastric cancer is the fifth most common cancer worldwide and remains the second leading cause of death (738,000 deaths annually in 2018) of all cancers in the world[8]. Despite advances in the early detection, diagnosis, and treatment of gastric cancer, the overall prognosis is still poor, and the 5-year survival for patients with gastric cancer has remained at 30%, which is due to recurrence and metastasis after surgery[9]. Moreover, the survival rate decreases to 5.2% in patients with distant metastases, who comprise 35% of the total number of gastric cancer patients. Thus, a better understanding of the molecular mechanisms underlying the metastatic process of gastric cancer is necessary to improve the treatment of gastric cancer and increase the survival rate. Considering that a major source of phenotypic plasticity that metastatic cells display is alternative splicing and that 30% more alternative splicing events were recently identified in 32 cancer types included in The Cancer Genome Atlas database (TCGA)[10], further comprehensive work is needed to identify the roles of spliced variants and splicing factors involved in the dysregulation of alternative splicing occurring specifically in gastric cancer development and metastasis.

In this study, we demonstrate an alternative splicing target of ESRP1, leucine-rich repeat Fli-1-interacting protein 2 (LRRFIP2), in gastric cancer cells. LRRFIP2 variant 2, an exon 7-truncated form of LRRFIP2 variant 3, was more highly expressed in ESRP1-high conditions, while LRRFIP2 variant 3 was more highly expressed in ESRP1-low conditions. Overexpression of LRRFIP2 variant 3 contributed to the metastasis of gastric cancer cells by modulating the histone methylation activity of coactivator-associated arginine methyltransferase 1 (CARM1). Taken together, we report LRRFIP2 as an alternative splicing target of ESRP1 in gastric cancer, which represents a regulatory mechanism of LRRFIP2 splicing variants in gastric cancer metastasis.

## Results

### The relative frequencies of *LRRFIP2* alternative splicing are significantly associated with the expression levels of *ESRP1* in human gastric cancer cell lines and gastric cancer patient tissues

To determine the clinical relevance of ESRP1 in gastric cancer, we first correlated the gene expression levels with the overall survival. As in the previous studies about ESRP1 in other cancers, high expression of *ESRP1* showed longer overall survival times (Supplementary Fig. 1)[11,12]. Based on the expression level of *ESRP1* in 18 gastric cancer cell lines, we divided them into two groups; *ESRP1*-low and -high (Fig. 1a). Interestingly, the majority of the cell lines in *ESRP1*-low group (SNU668, SNU484, and MKN1) was found to be mesenchymal subtype, while the majority of the cell lines in *ESRP1*-high group (SNU638, SNU719, KATOIII, AGS, SNU601, SNU620, MKN45, NCI-N87, MKN74, SNU216, MKN28, and SNU16) was epithelial subtype (Supplementary Fig. 2)[13], suggesting that ESRP1 expression may be involved in the determination of epithelial/mesenchymal phenotype of gastric cancer cells.

To investigate the impact of ESRP1 on the mRNA splicing patterns of gastric cancer cell lines, our previously published RNA sequencing data[14] from 18 gastric cancer cell lines expressing varying levels of *ESRP1* were reanalyzed. We performed isoform switch analysis using iso-KTSP[15,16], which revealed 100 genes for which the relative frequency of splicing isoforms differed in *ESRP1*-low versus *ESRP1*-high cell lines 20 candidate genes with high correlation scores are shown in the heatmaps (Fig. 1b, c and Supplementary Table 4). As in the RNA sequencing data of the gastric cancer cell lines, the tissue samples were arranged in ascending order of *ESRP1* expression, and the differential patterns of the relative frequencies of splicing isoforms were analyzed and the top 20 candidate genes with high correlation scores are shown

in the heatmaps (Supplementary Fig. 3a–c). We also investigated splicing event types using SUPPA[17] (Supplementary Table 5–11).

To examine the biological processes altered by ESRP1, we identified gene ontology (GO) terms and Kyoto Encyclopedia of Genes and Genomes (KEGG) pathways using the DAVID functional annotation tool. The results suggested that the upregulated genes in *ESRP1*-low cell lines were highly enriched for terms and pathways such as basal cell carcinoma, negative regulation of epithelial cell differentiation, and extracellular matrix organization (Fig. 1d). Conversely, upregulated genes in *ESRP1*-high cell lines were related to KEGG pathways, such as adherent and tight junctions, and to GO terms, such as positive regulation of cell adhesion, regulation of motility and cell division, and cell-cell junction assembly (Fig. 1e). The genes upregulated in the *ESRP1*-low condition in the gastric tumor tissues were highly enriched in GO terms and KEGG pathways, including focal adhesion, regulation of cell migration, and cell junction assembly (Supplementary Fig. 3d). Conversely, the genes upregulated in the *ESRP1*-high condition were highly enriched in GO terms and KEGG pathways, including tight junction and cell-cell junction organization (Supplementary Fig. 3e). Although there is some discrepancy in the terms and pathways between cell lines and tumor tissues due to the heterogenic characteristic of gastric cancer tissues, the terms and pathways regarding cell junctions, proliferation and motility show consistency, supporting the notion that ESRP1 has a critical function in metastasis and tumorigenesis by suppressing tumor motility and invasiveness in gastric cancer cells.

Among the top candidate genes, *LRRFIP2* was found to be an as yet unidentified alternative splicing candidate whose splicing event was highly correlated with *ESRP1* expression (Supplementary Table 4). Under *ESRP1*-low conditions, exon 7 of *LRRFIP2*_NM_00134369 (referred hereafter as *LRRFIP2* variant 3) was intact, while the exon was skipped under *ESRP1*-high conditions (*LRRFIP2*_NM_017724; referred hereafter as *LRRFIP2* variant 2) (Fig. 1f, h and Supplementary Fig. 3f, g). In addition to *LRRFIP2*, *CCDC50*, another splicing candidate of ESRP1, also showed skipping exon event (Supplementary Table 11) and BICD2 showed retained intron event (Supplementary Table 10) occurring in gastric cancer cells depending on the expression of *ESRP1* (Supplementary Fig. 4).

The *LRRFIP2* gene can generate four mRNA transcripts through pre-mRNA alternative splicing (Supplementary Fig. 5a). However, only *LRRFIP2* variant 2 and *LRRFIP2* variant 3 showed significantly distinct expression patterns in association with *ESRP1*: higher expression of *LRRFIP2* variant 2 in *ESRP1*-high cell lines and *LRRFIP2* variant 3 in *ESRP1*-low cell lines ($P < 0.0001$) (Supplementary Fig. 5b). The frequencies of *LRRFIP2* splicing events were directly correlated with the *ESRP1* expression level, and the RNA sequencing results from the 18 gastric cancer cell lines and 18 tissue samples were confirmed by RT-PCR (Fig. 1i, j and Supplementary Fig. 3h). These observations support the notion that isoform switching event of *LRRFIP2* may be critical for determination of the epithelial phenotype regulated by ESRP1 in gastric cancer cells.

An interrogation of the gastric cancer data sets in TCGA was performed to examine the correlation between the expression of *ESRP1* and *LRRFIP2* variant 3 in metastatic progression. Of note, a significantly lower expression level of *ESRP1* and a higher expression level of *LRRFIP2* variant 3 were observed in more advanced gastric cancer stages (Fig. 1k, l). Consistently, analysis by metastatic stage demonstrated that the cases with metastatic gastric cancer had significantly lower mRNA expression of *ESRP1* and higher mRNA expression of *LRRFIP2* variant 3 (Fig. 1m, n). To address the functional significance of the association of *LRRFIP2* variant 3 expression with clinical outcomes in gastric cancers, we performed Kaplan-Meier survival using gene expression profiles and clinical data from stage IV gastric cancer patients[18,19]. Notably, gastric cancer patients with higher than twofold upregulated *LRRFIP2* variant 3 expression in tumor tissues compared to their matched normal tissues exhibited significantly shorter overall

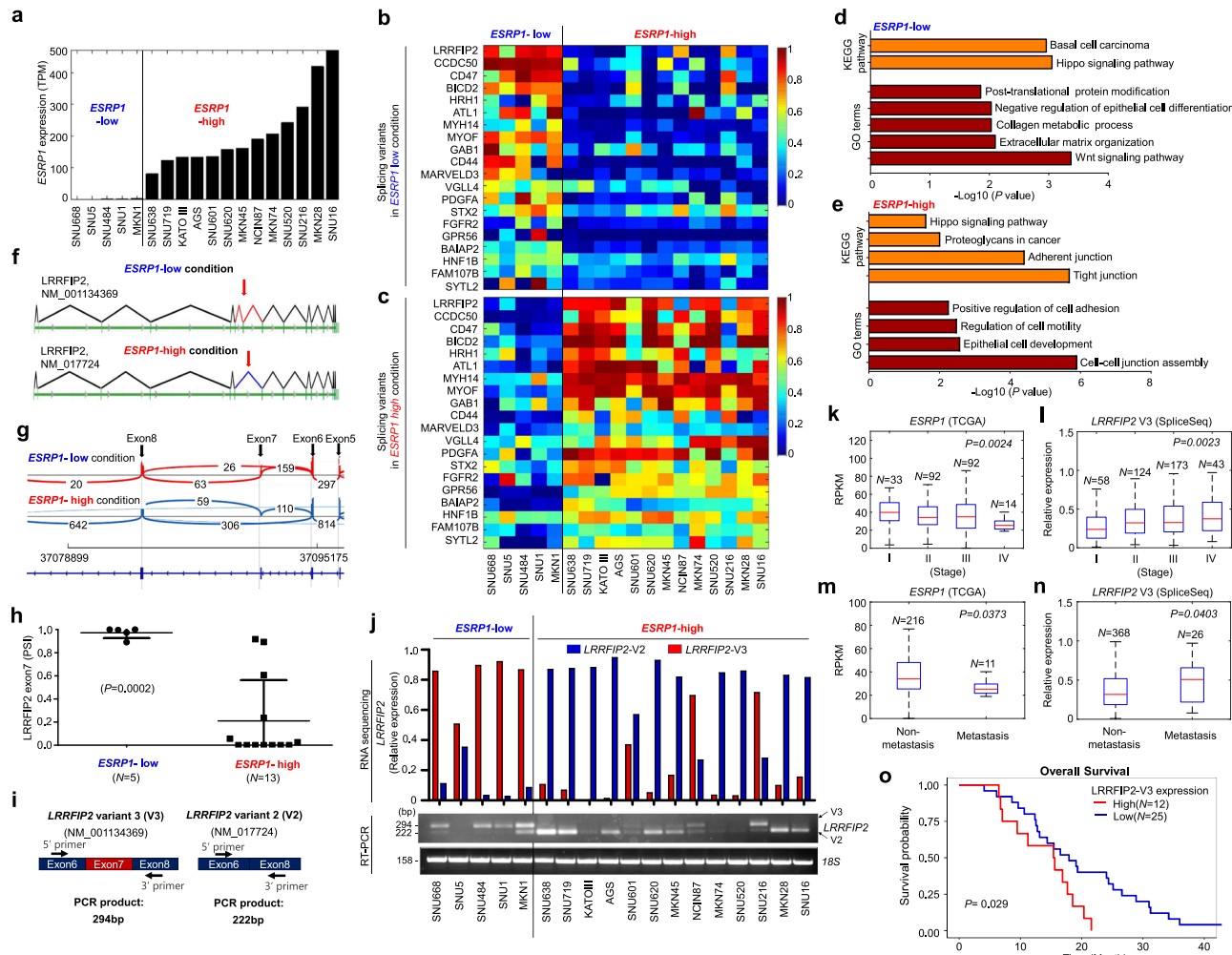

**Fig. 1 | The relative frequencies of *LRRFIP2* alternative splicing are significantly associated with the expression levels of *ESRP1* in human gastric cancer cell lines and gastric cancer patient tissues. a** Bar graph showing the expression levels of *ESRP1* differentially expressed in the gastric cancer cell lines. **b**, **c** Heatmaps illustrating relative TPM of 20 splicing variants alternatively expressed in **b** *ESRP1*-low and **c** -high conditions. **d**, **e** KEGG pathways and GO terms enriched in differentially expressed genes upregulated in *ESRP1*-low cell lines and in *ESRP1*-high cell lines from **b**, **c**. **f** Schematic representation of human *LRRFIP2* isoforms differentially expressed in *ESRP1*-low condition and *ESRP1*-high condition. **g** Sashimi plots indicate exon usage and splicing of *LRRFIP2* in *ESRP1*-low and -high conditions. Arches and numbers represent RNA-seq reads at exon–exon junctions. **h** Dot plot derived from PSI of *LRRFIP2* exon 7 in the gastric cancer cell lines. Data are representative mean PSI ± SD of individual cell lines in *ESRP1*-low group (*N* = 5) and *ESRP1*-high group (*N* = 13). **i** Schematic representation of PCR analysis of *LRRFIP2* variants using the same set of primers. **j** Bar graph and RT-PCR result showing expression levels of

*LRRFIP2* variant 2 (shown in red) and variant 3 (shown in blue) in 18 gastric cancer cell lines. **k**–**n** Box plots showing *ESRP1* and *LRRFIP2* variant 3 expression levels in gastric cancer tissues from public TCGA data sets, **k**, **l** in cancer stages I-IV and **m**, **n** in non-metastatic versus metastatic gastric cancers. The center line is the median; the box is from the 25th to the 75th percentile. The upper or lower whisker extends from the hinge to the 1.5 x IQR (distance between the first and third quartiles) from the hinge for up and low, respectively. **o** A plot showing Kaplan-Meier curves which represent the overall survival differences between patients who had highly up-regulated *LRRFIP2* variant 3 (*LRRFIP2*-V3) expression levels in tumors compared to their matched normal tissues (≥2-fold) (*N* = 12) and other patients (*N* = 25) in the stage IV gastric cancer dataset. Overall survival was defined as the time (months) from the diagnosis date of stage IV to death. *P* value for **h** was calculated by unpaired two-tailed Student's t tests. *P* values for **k**–**o** were calculated from the log-rank test. Source data are provided in the Source Data file.

survival times than others (Fig. 1o). Thus, these data demonstrate that the expression of *LRRFIP2* variant 3 is negatively correlated with *ESRP1* expression and positively correlated with metastatic potential in gastric tumor tissues.

Taken together, these results suggest that the relative frequencies of *LRRFIP2* isoform switching are highly correlated with the expression levels of *ESRP1* in human gastric cancer cell lines and gastric patient tissues.

### Overexpression of LRRFIP2 variant 3 increases the metastatic potential of gastric cancer cells, while deletion of exon 7 in LRRFIP2 decreases it

We next examined the effect of ESRP1 regulation on *LRRFIP2* variant changes in gastric cancer. Studies have reported that ESRP1 plays a

central role in suppressing tumorigenic potential and/or attenuating metastasis in various types of cancer[2,5,20]. To identify the role of ESRP1 in regulating the invasiveness and motility of gastric cancer cells, we conducted transwell migration and Matrigel invasion assays using MKN1 cells, which have low basal expression of ESRP1, following ectopic overexpression of ESRP1. Overexpression of ESRP1 significantly decreased the migration and invasion of MKN1 cells (Fig. 2a), while it had no significant effect on their proliferation or foci-forming ability (Supplementary Fig. 6a, b). To further examine whether the regulation of the alternative splicing of *LRRFIP2* by ESRP1 is a direct event, we analyzed the expression of *LRRFIP2* variants in MKN1 and SNU484 cells stably overexpressing ESRP1. Consistent with the previous results, ectopic overexpression of ESRP1 decreased the expression of *LRRFIP2* variant 3, which is otherwise mainly expressed in MKN1

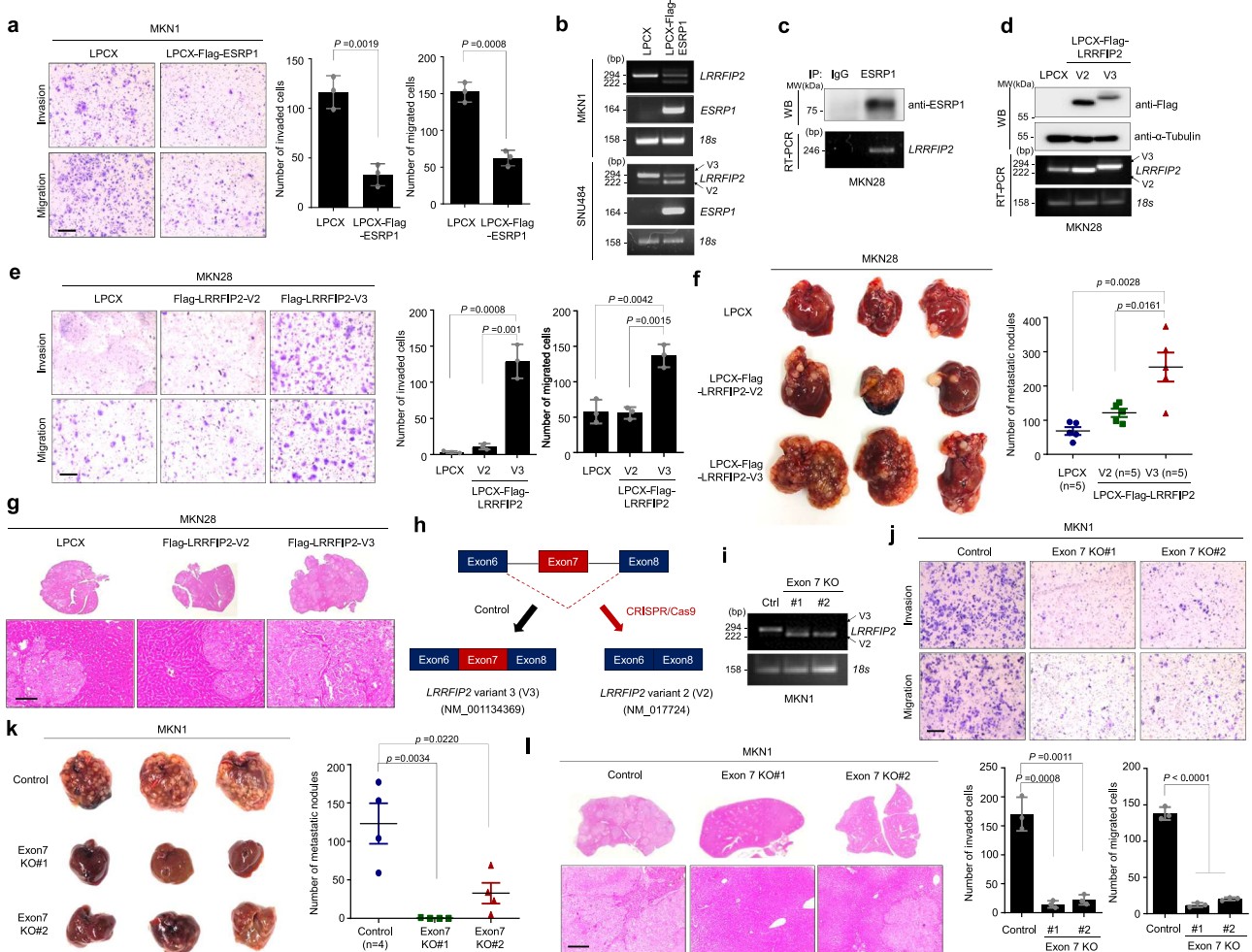

**Fig. 2 | Exon 7 of LRRFIP2 variant 3 determines the metastatic potential of gastric cancer cells in vitro and in vivo. a** Transwell migration assay and Matrigel invasion assay of MKN1 cell lines stably expressing ESRP1 proteins (left) and bar graphs showing number of invaded and migrated cells (right), respectively, following staining with crystal violet. Original magnification, _40X. Scale bar, 0.5 mm. **b** RT-PCR analysis showing overexpression of *ESRP1* and *LRRFIP2* variant 2 and 3. **c** Immunoblot and RT-PCR analysis of ESRP1 immunoprecipitation. **d** Immunoblot and RT-PCR analysis of LPCX-Flag-LRRFIP2 variants 2 and 3-overexpressing MKN28 cells. **e** Transwell migration assay and Matrigel invasion assay of MKN28 cell lines stably expressing LRRFIP2 variant 2 and 3 proteins (left) and bar graphs showing number of invaded and migrated cells (right), respectively, following staining with crystal violet. Original magnification, _40X. Scale bar, 0.5 mm. **f** Representative whole liver image showing metastatic nodules (left) and scatter plot showing the number of liver metastatic nodules (right). **g** H&E staining showing sections of the metastasized liver from **f**. Original magnification, _100X. Scale bar, 100 μm.

**h** Schematic representation of exon 7 deletion by CRISPR/Cas9 system. **i** RT-PCR analysis of *LRRFIP2* in exon 7-deleted cell lines. **j** Transwell migration assay and Matrigel invasion assay of exon 7-deleted cell lines (top) and bar graphs showing number of invaded and migrated cells (bottom), respectively, following staining with crystal violet. Original magnification, _40X. Scale bar, 0.5 mm.
**k** Representative whole liver image showing metastatic nodules (left) and scatter plot showing the number of liver metastatic nodules (right). **l** H&E staining showing sections of the metastasized liver from **k**. Original magnification, _100X. Scale bar, 100 μm. **a**, **e**, **j** Data are representative mean ± SD of three independent experiments (*N* = 3). **b**–**d**, **i** The representative results were obtained from at least three independent experiments. **f** Data are representative mean ± SD of five independent animals (*n* = 5). **f** Data are representative mean ± SD of four independent animals (*n* = 4). All *P* values were calculated by unpaired two-tailed Student's t tests. These data represent the mean ± S.D. Source data are provided in the Source Data file.

and SNU484 cells, while it increased the expression of variant 2 (Fig. 2b). As recent studies demonstrate that ESRP binding motifs are often observed upstream of ESRP-silenced exons[21], we screened several regions upstream and downstream of *LRRFIP2* V3 exon 7 to reveal the binding of ESRP1 to upstream of exon 7 (Fig. 2c and Supplementary Fig. 7). Collectively, our results suggest that a direct interaction between the ESRP1 and *LRRFIP2* gene induces silencing of exon 7 of *LRRFIP2* variant 3.

Having evidence of ESRP1 regulation of *LRRFIP2* variant changes, we then evaluated the physiological functions of the two variants of *LRRFIP2* in gastric cancer progression and metastasis by generating cell lines stably overexpressing LRRFIP2 variants 2 and 3 (Fig. 2d). In particular, overexpression of LRRFIP2 variant 3 significantly increased the cell migration and invasion ability of MKN28 and

MKN74 cells with relatively high ESRP1 expression, but no significant change was observed due to overexpression of LRRFIP2 variant 2 (Fig. 2e and Supplementary Fig. 8a). Using MKN28 cell lines which are often used as an in vivo model for liver metastasis, we further observed that only LRRFIP2 variant 3-overexpression led to a significant induction of liver metastasis in immunodeficient mice (Fig. 2f, g)[22,23]. Next, we evaluated whether overexpression of LRRFIP2 variants 2 and 3 affected cell proliferation and tumor growth. Interestingly, the isoform switch did not have much effect on the proliferation of the gastric cancer cell line in vitro (Supplementary Fig. 6c). To confirm that this result can also be observed in vivo, we injected MKN28 cells overexpressing LRRFIP2 variants 2 and 3 subcutaneously into the flanks of immunodeficient mice. Consistent with the in vitro observations, neither of the two variants induced a

significant change in the tumor volume of gastric cancer cells (Supplementary Fig. 6d).

To further investigate whether the presence of exon 7 of LRRFIP2 variant 3 is indeed important for dramatic changes in metastatic potential in gastric cancer cells, we deleted exon 7 of endogenous *LRRFIP2* variant 3 using the CRISPR/Cas9 system. Here, we used MKN1 cells with low expression of *ESRP1* and high expression of variant 3 (Fig. 2h, i and Supplementary Fig. 9). We then tested the cell migration and invasion ability of LRRFIP2 exon 7-depleted MKN1 cells. Knockout of the exon dramatically reduced the migration and invasion of MKN1 cells, and also in SNU484 cells, while it had no significant effect on cell proliferation (Fig. 2j and Supplementary Figs. 6e and 10a). We overexpressed LRRFIP2 variant 3 in exon 7 knockout clones and investigated the rescue effect of the variant 3 to overcome the limitation of targeting multiple sites, as we ony had to knockout exon 7, not the entire LRRFIP2 gene. Exogenous introduction of variant 3, excluding potential off-target effects, restored reduced invasiveness and migratory potential (Supplementary Fig. 11a). Furthermore, we investigated the phenotypic change in vivo by spleen injection of LRRFIP2 exon 7-depleted MKN1 cells into immunodeficient mice and observed significant reductions in liver metastasis in both of the exon 7 knockout clones (Fig. 2k, l). Although the possibility that this phenotype is caused by the gain of function of LRRFIP2 variant 2 in addition to the loss of function of variant 3 cannot be ignored, we observed that exon 7 is indeed an important factor determining the metastatic properties of gastric cancer cells. Collectively, these in vitro and in vivo data suggest that the presence of exon 7 in LRRFIP2 variant 3 greatly increases the metastatic potential of gastric cancer cells.

### Exon 7 deletion in LRRFIP2 variant 3 induces changes in gene expression patterns

Given that the expression of LRRFIP2 variant 3 significantly induced the cell motility and invasiveness of gastric cancer cells, it seemed necessary to verify the functions of the isoforms of LRRFIP2 in regulating metastatic potential. Thus, we performed RNA sequencing of wild-type and exon 7-deleted MKN1 cells. Intriguingly, the exclusion of exon 7 exhibited significant changes in the expression levels of a number of genes, as shown in the heatmap (with a twofold cutoff, $P < 0.001$) (Fig. 3a). To gain further insights into the genes up- or downregulated by the isoform switch, we conducted GO and KEGG pathway analyses using the DAVID functional annotation tool and observed that the downregulated genes in LRRFIP2 exon 7-depleted cells were highly enriched for pathways such as cell adhesion, cell migration, and extracellular matrix organization (Fig. 3b). Then, we confirmed some of the up- or downregulated genes that were reported to have significant roles in cancer cells exhibiting an aggressive phenotype[24–29] by quantitative RT-PCR (Fig. 3c and Supplementary Fig. 12a). Furthermore, to examine whether these genes were also upregulated in LRRFIP2 variant 3-transduced cell lines, we conducted quantitative RT-PCR using MKN28 cells overexpressing LRRFIP2 variants 2 and 3. Notably, the genes that were downregulated in exon 7-depleted MKN1 cells were upregulated by overexpressing LRRFIP2 variant 3 in MKN28 cells (Fig. 3d and Supplementary Fig. 12b).

Additionally, to investigate the clinical relevance of the target gene expression in gastric cancer, we examined their expression levels at different stages of TCGA gastric cancer samples and observed significantly higher expression of *SERPINE1, COL5A2, SEMA3C* and *LOXL2* in more aggressive stages (Fig. 3e and Supplementary Fig. 12c). To verify the functional significance of the association of some of these genes with clinical outcomes in gastric cancer, we performed public meta-analyses using Kaplan-Meier Plotter software. Of note, patients with high expression of *SERPINE1, LOXL2*, and *CDK6* displayed significantly shorter relapse-free survival than those with low expression (Fig. 3f and Supplementary Fig. 12d). Taken together, these results show that distinct LRRFIP2 variants can differentially modulate the

transcription of genes essential for metastasis, which may underlie the phenotypes observed in vitro and in vivo.

### CARM1 co-activates transcription of *SERPINE1* in conjunction with LRRFIP2 variant 3 in gastric cancer cells

To explore the molecular mechanism underlying the differentially regulated gene transcription in cells with altered relative frequencies of LRRFIP2 variants, we performed a mass spectrometry analysis using LRRFIP2 variant 2- and 3-overexpressing MKN28 cells to identify an interacting partner. Mass spectrometry analysis suggested coactivator-associated arginine methyltransferase 1 (CARM1), a type I protein arginine methyltransferase (PRMT) that acts as a transcriptional coactivator by asymmetrically dimethylating protein substrates on arginine residues, as a potential binding protein of LRRFIP2 variant 2 exclusively (Fig. 4a). Emerging evidence suggests that CARM1 functions as an oncogene in human cancers, and it is often highly expressed in several cancer types, such as breast, colon, and prostate cancers[30–33]. The mRNA expression of *CARM1* was also significantly increased in gastric cancer tissues compared to normal tissues (Supplementary Fig. 13a). Public meta-analyses using Kaplan-Meier Plotter software also demonstrated that patients with high *CARM1* expression displayed significantly shorter relapse-free survival times than those with low expression (Supplementary Fig. 13b). An immunoprecipitation assay revealed that CARM1 exhibited a strong interaction with LRRFIP2 variant 2 but a weak interaction with variant 3 (Fig. 4b, c). To further support our mass-spectrometry data and immunoprecipitation data, we examined the intracellular localizations of CARM1 and LRRFIP2. Both LRRFIP2 and CARM1 were detected in the nuclear fraction. However CARM1 was found only in the nucleus, whereas LRRFIP2 was found both in the cytoplasm and nucleus (Supplementary Fig. 14).

As an epigenetic regulator, CARM1 exerts its transcriptional control by methylating arginine residues on histones with specificity for histone H3R17, transcription factors, and other co-transcriptional regulators[34]. We observed that knockdown of CARM1 decreased the asymmetric dimethylation of H3R17 in gastric cancer cells, as expected (Fig. 4d). However, no significant change in BAF155 R1064 dimethylation, which is another well-known target of CARM1 in breast cancer cells[35], was observed, suggesting that dimethylation of BAF155 R1064 might be cell type-dependent (Supplementary Fig. 15a). To investigate the roles of LRRFIP2 variants in CARM1-mediated methylation and the subsequent transcriptional regulation of its target genes, we first assessed the dimethylation status of H3R17 when LRRFIP2 variants 2 and 3 were overexpressed in MKN28 cells, and exon 7 of endogenous LRRFIP2 variant 3 was eliminated in MKN1 cells. Intriguingly, H3R17 dimethylation was significantly enhanced only when LRRFIP2 variant 3 was ectopically overexpressed in MKN28 and MKN74 cells and was reduced when exon 7 of LRRFIP2 was deleted in MKN1 and SNU484 cells (Fig. 4e, f and Supplementary Figs. 8b and 10b). Furthermore, overexpression of LRRFIP2 variant 3 in the exon 7 knockout cell line rescued methylation of H3R17 by CARM1 (Supplementary Fig. 11b). Conversely, no significant change in BAF155 R1064 dimethylation was observed by the overexpression of LRRFIP2 variants or exon 7 deletion, consistently suggesting that CARM1 alone or in conjunction with LRRFIP2 might not be crucial in the regulation of BAF155 R1064 dimethylation in gastric cancer cells (Supplementary Fig. 15b, c).

These findings led us to examine whether elevated dimethylation of H3R17 by CARM1 in the presence of LRRFIP2 variant 3 resulted in the induction of gene transcription. Among the genes identified from RNA sequencing (Fig. 3a), *SERPINE1* has been shown to play a key role in tumor metastases[24,29], often leading to poor prognosis in various cancers. As we observed that the mRNA expression and promoter activity of *SERPINE1* were significantly regulated by the isoform switch of LRRFIP2 (Fig. 3c, d and Supplementary Figs. 8c, 10c, 11c–d and 16a), we assessed the functional relevance of these LRRFIP2 splicing variants

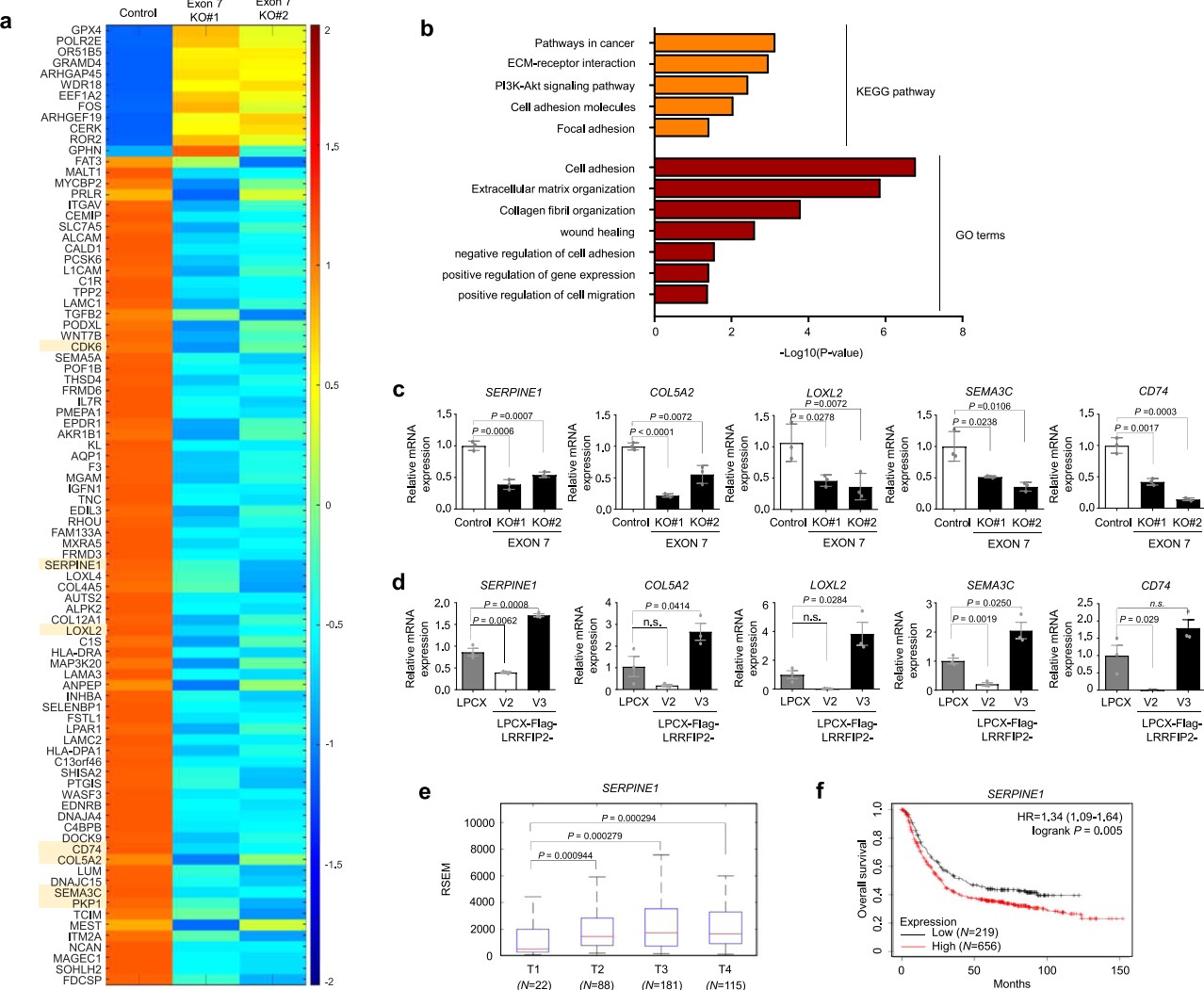

**Fig. 3 | Alternative splicing of LRRFIP2 regulates metastasis-associated gene network in gastric cancer cells. a** Heatmap showing the downregulated genes in exon 7 of *LRRFIP2* variant 3-deleted MKN1 cells. Threshold values are as follows: corrected value *P* < 0.01 and absolute log2-fold change >1.0. **b** KEGG pathways and GO terms enriched in differentially expressed genes downregulated by exon 7 deletion from **a**. **c** Real-time qRT-PCR showing the expression of altered target genes in exon 7-deleted MKN1 cells. **d** Real-time qRT-PCR showing the expression of the altered target genes from **c** in MKN28 cells stably expressing *LRRFIP2* variant 2 and 3. **e** Box plots showing *SERPINE1* expression levels in each tumor stage in gastric cancer tissues from public TCGA data sets. T1-T4 refers to the size and or/extent of

the main tumor into nearby tissues. The center line is the median; the box is from the 25th to the 75th percentile. The upper or lower whisker extends from the hinge to the 1.5 x IQR (distance between the first and third quartiles) from the hinge for up and low, respectively. **f** Kaplan-Meier analysis showing relapse-free survival depending on *SERPINE1* expression levels from public meta-analysis data (*N* = 875). *P* values for **c**, **d** were calculated by unpaired two-tailed Student's t tests. *P* values for **e** were calculated by two-sided Wilcoxon rank sum tests. *P* value for **f** was calculated from log-rank test. **c**, **d** Data are representative mean ± SD of three independent samples (*N* = 3). n.s: not significant. Source data are provided in the Source Data file.

for *SERPINE1* gene expression in conjunction with CARM1. The upregulated mRNA expression of *SERPINE1* in LRRFIP2 variant 3-overexpressing MKN28 cells was reduced by the knockdown of CARM1 (Fig. 4g), suggesting that SERPINE1 might be transcriptionally activated by CARM1 in the presence of LRRFIP2 variant 3. In addition, the promoter activity of *SERPINE1* was downregulated by the knockdown of CARM1 (Supplementary Fig. 16b).

Furthermore, Chromatin immunoprecipitation (ChIP) assays were performed on cells overexpressing LRRFIP2 variants 2 and 3 with antibodies against human CARM1 and H3R17me2 and showed that the CARM1 protein and asymmetrically dimethylated H3R17 protein were present at the human *SERPINE1* promoter in the presence of LRRFIP2 variant 3 (Fig. 4h, i). Interestingly, when the expression of the SERPINE1 protein and dimethylated H3R17 protein was examined in metastatic liver tumor tissues, it was observed that the expression of these proteins were significantly reduced in MKN1 cells with LRRFIP2 exon 7

deletion compared to control MKN1 cells (Figs. 3k and 4j), suggesting that both CARM1 and LRRFIP2 variant 3 are involved in the regulation of the expression of *SERPINE1*. Additionally, induction of CARM1 recruitment to the *CCNE1* promoter, a known CARM1 target, was also observed in the presence of LRRFIP2 variant 3 by ChIP assay, suggesting that transcriptional regulation by CARM1 in other CARM-targeted genes may be associated with the alternative splicing of *LRRFIP2* (Supplementary Fig. 17a). Indeed, we observed elevated expression levels of the known target genes of CARM1, including *AXIN2*, *CCNE1*, and *GADD45A*, in MKN28 cells overexpressing LRRFIP2 variant 3, while they were not upregulated in variant 2-overexpressing cells (Supplementary Fig. 17b).

Furthermore, to validate whether alteration of SERPINE1 expression and asymmetric dimethylation of H3R17 by alternative splicing of LRRFIP2 can be observed in vivo, we performed immunohistochemical analyses using anti-SERPINE1 and anti-H3R17me2a antibodies on the

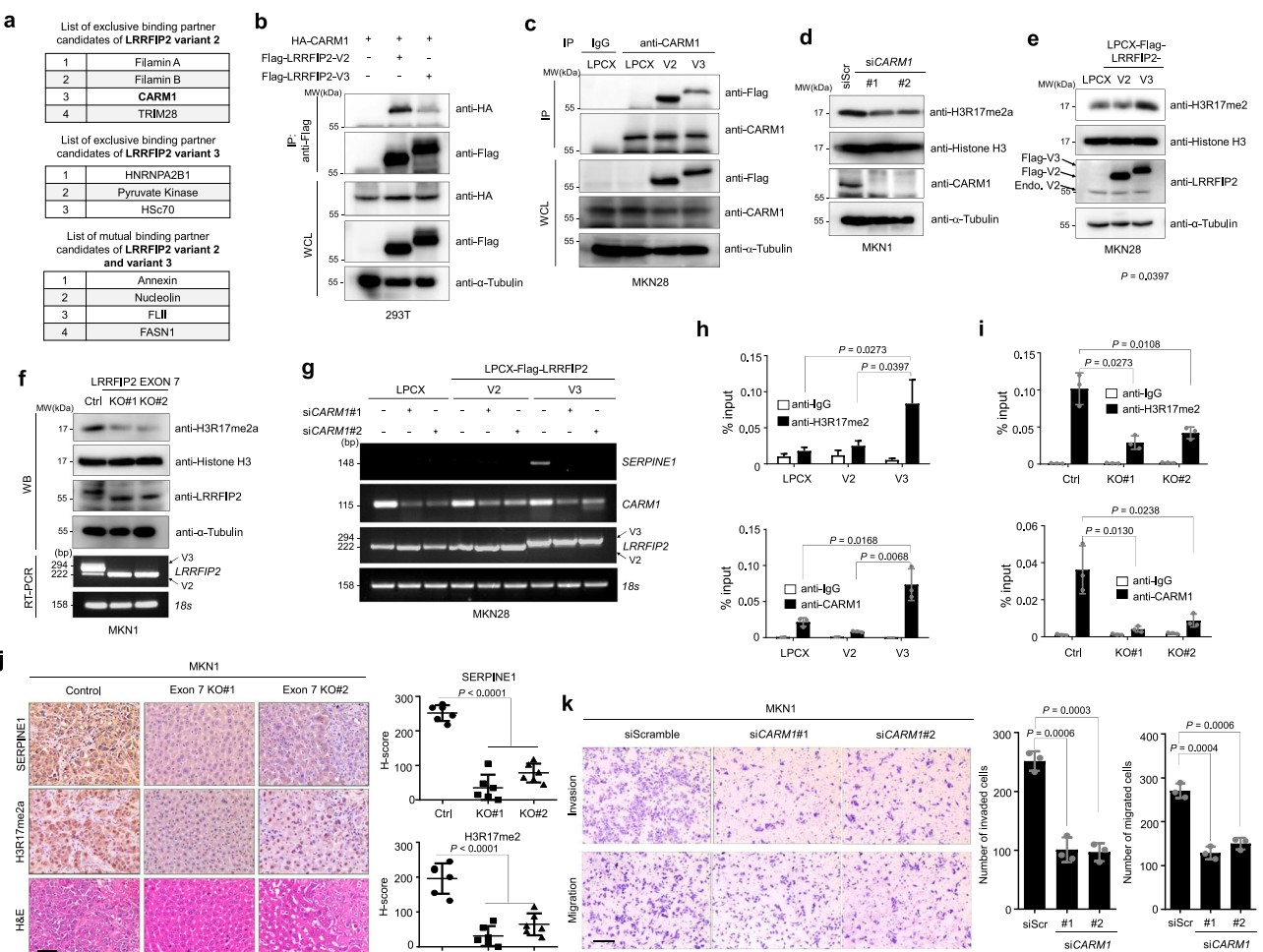

**Fig. 4 | Alternative splicing of LRRFIP2 regulates histone methyltransferase activity of CARM1 through specific interaction. a** List of LRRFIP2 binding partner candidates analyzed from the mass spectrometry-base crosslinking assay. **b** Immunoprecipitation assay showing the interaction between HA-CARM1 and Flag-LRRFIP2 variant 2 or 3 in 293 T cells. **c** Immunoprecipitation assay showing the interaction between LRRFIP2 variant 2 or 3 and endogenous CARM1. **d** Immunoblot analysis showing H3R17me2a, histone H3, and CARM1 in control and CARM1 siRNA-treated cells. **e** Immunoblot analysis showing H3R17me2a, histone H3 and Flag in LRRFIP2 variant 2 and 3-overexpressing MKN28 cells. **f** Immunoblot and RT-PCR analysis showing H3R17me2a, histone H3 and *LRRFIP2* in control and exon 7-deleted cell lines. **g** RT-PCR analysis showing expression of *SERPINE1, CARM1, LRRFIP2* in LRRFIP2 variant 2 or 3-overexpressing MKN28 cells, with CARM1 knockdown by siRNA transfection. **h, i** qChIP analysis of the human *SERPINE1* promoter with antibodies to CARM1, H3R17me2a in **h** LRRFIP2 variant 2 and 3-overexpressing

MKN28 cell lines and **i** LRRFIP2 variant 3 exon7 deleted MKN1 cell lines. **j** Representative IHC images showing SERPINE1 and H3R17me2a expression in metastasized liver tissues from Fig. 2k (left) and H-score scatter plots of the IHC staining (right). Original magnification, _200X. Scale bar, 50 μm. **k** Transwell migration assay and Matrigel invasion assay of MKN1 cells with CARM1 knockdown (left) and bar graphs showing number of invaded and migrated cells (right), respectively, following staining with crystal violet. Original magnification, _40X. Scale bar, 0.5 mm. **b–g** The representative results were obtained from at least three independent experiments. **h, i** Data are representative mean ± SD of three independent samples (*N* = 3). **j** Data are representative mean ± SD of six independent samples (*n* = 6). **k** Data are representative mean ± SD of three independent experiments (*N* = 3). All *P* values were calculated by unpaired two-tailed Student's t tests. Source data are provided in the Source Data file.

metastasized liver tissue, as shown in Fig. 2k, l. Consistent with our previous results, the expression of SERPINE1 and the dimethylation of H3R17 were significantly decreased in the liver tissues of the mice injected with LRRFIP2 exon 7-deleted cells (Fig. 4j). Considering that CARM1 expression is critical for SERPINE1 expression, we also examined whether CARM1 actually affected the migration and invasion of gastric cancer cells. Indeed, knockdown of CARM1 significantly decreased the migratory and invasive capacities of MKN1 cells (Fig. 4k). Interestingly, LRRFIP2 variant 3 was shown to bind LRRFIP2 variant 2 and inhibit the binding of LRRFIP2 variant 2 to CARM1 (Supplementary Fig. 18), suggesting that LRRFIP2 variant 3 suppresses tumor suppressor activity of LRRFIP2 variant 2 by blocking the interaction between CARM1 and LRRFIP2 variant 2. Taken together, LRRFIP2 variant 3 may assist the recruitment of CARM1 to target genes, regulating the transcriptional activation of metastasis-promoting genes such as *SERPINE1*

## CARM1 interacts with ACTR in the transcriptional regulation of *SERPINE1* in the presence of LRRFIP2 variant 3

CARM1 was originally identified through its binding to GRIP1/TIF2/Src-2/NCOA2, a member of the p160 family of steroid receptor coactivators, in a yeast two-hybrid screen, and other p160 family members (Src-1/NCOA1, ACTR/AIB1/SRC-3/NCOA3) were also shown to directly interact with CARM1[36–39]. Since the p160 coactivator family serves as a binding platform for CARM1, assisting its role as a coregulator of transcription, we investigated whether LRRFIP2 variant 2 diminishes the activity of CARM1 by abrogating its interaction with a member of the p160 family. Interestingly, exon 7 deletion led to reduced interaction between ACTR and CARM1 (Fig. 5a). Consistently, LRRFIP2 variant 3 overexpression increased the interaction between CARM1 and ACTR, whereas overexpression of LRRFIP2 variant 2 slightly reduced interaction between CARM1 and ACTR in MKN28 cells (Fig. 5b). To further determine whether LRRFIP2 variant 2 interferes

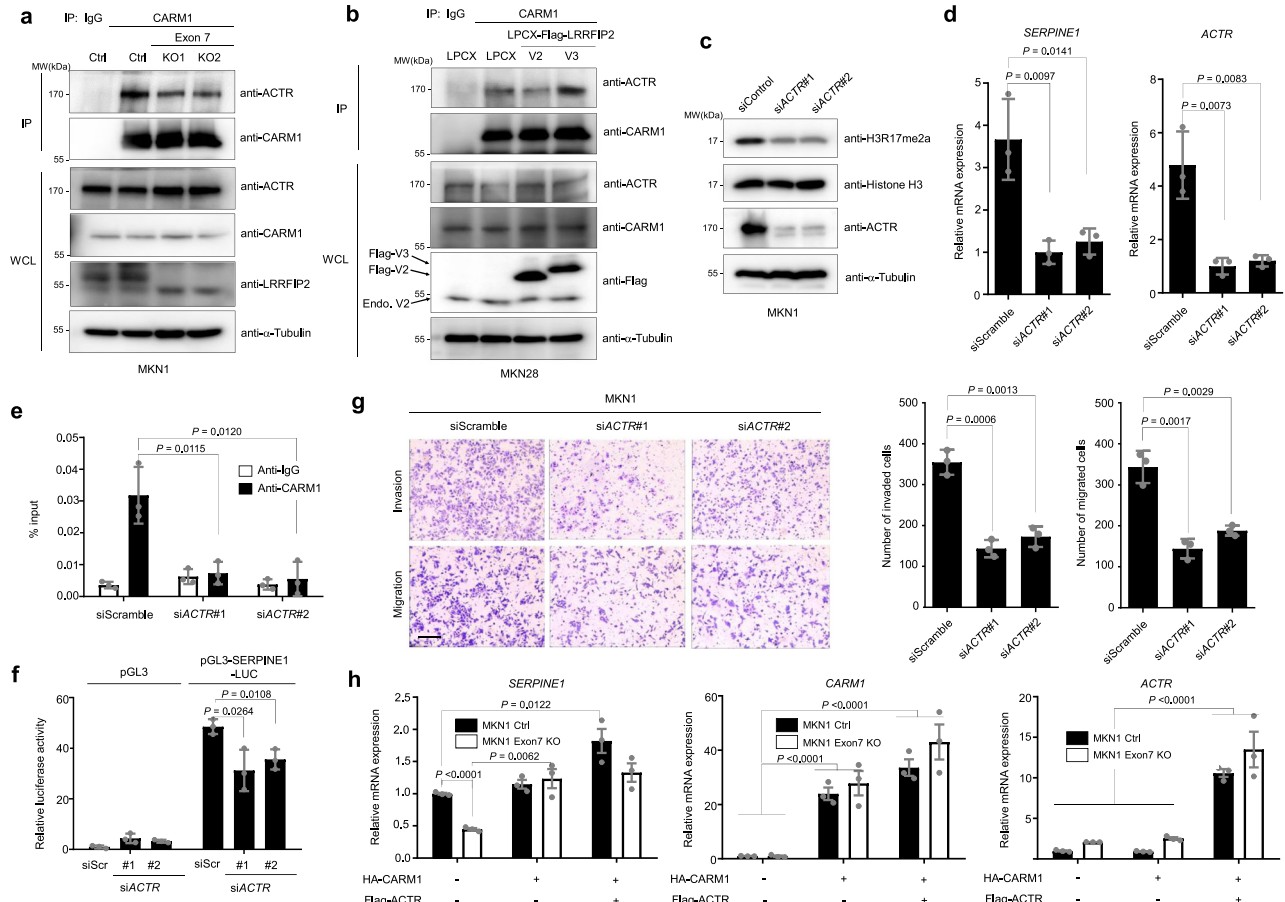

**Fig. 5 | CARM1 and the p160 coactivator ACTR/SRC3/AIB1/NCOA3 cooperate for transcriptional regulation of *SERPINE1*. a** Immunoprecipitation assay showing the interaction between endogenous CARM1 and members of ACTR in control and exon7-deleted MKN1 cell lines and RT-PCR analysis showing expression of *LRRFIP2* variants. **b** Immunoprecipitation assay showing the interaction between endogenous CARM1 and members of ACTR in LRRFIP2 variants 2 and 3-overexpressing MKN28 cell lines. **c** Immunoblot analysis showing expression of H3R17me2a and ACTR following knockdown of ACTR in MKN1 cells. **d** RT-PCR analysis showing expression of *SERPINE1* and *ACTR* following knockdown of ACTR in MKN1 cells. **e** qChIP analysis of the human *SERPINE1* promoter with antibodies to CARM1, H3R17me2a in MKN1 cells following control and *ACTR* siRNA transfection. **f** Control

and ACTR-knockdown MKN1 cells were transfected with *SERPINE1* promoter (−1500/+500) and then assayed for luciferase activity. **g** Transwell migration assay and Matrigel invasion assay of MKN1 cells following knockdown of ACTR (left) and bar graphs showing number of invaded and migrated cells (right), respectively, following staining with crystal violet. Original magnification, _40X. Scale bar, 0.5 mm. **h** qRT-PCR result showing the relative mRNA expression of *SERPINE1*, *CARM1*, and *ACTR*. All *P* values were calculated by unpaired two-tailed Student's t tests. **a**–**c** The representative results were obtained from at least three independent experiments. **d**–**f**, **h** Data are representative mean ± SD of three independent samples (*N* = 3). **g** Data are representative mean ± SD of three independent experiments (*N* = 3). Source data are provided in the Source Data file.

the interaction between ACTR and CARM1 through direct binding to CARM1, we generated a cell line overexpressing LRRFIP2 variant 2 in MKN1 cells with high expression of LRRFIP2 variant 3. Interestingly, the binding of ACTR to CARM1 was markedly reduced by ectopic expression of LRRFIP2 variant 2 (Supplementary Fig. 19a), suggesting that direct binding of variant 2 to CARM1 prevents ACTR from binding to CARM1. In addition, overexpression of LRRFIP2 variant 2 in MKN1 cells downregulated the expression of *SERPINE1*, asymmetric dimethylation of histone H3R17, invasiveness and migratory potential (Supplementary Fig. 19b–d).

Then, we investigated the role of ACTR in CARM1-mediated gene transcription in MKN1 cells predominantly expressing LRRFIP2 variant 3. ACTR knockdown reduced the asymmetric dimethylation of histone H3R17 in gastric cancer cells, suggesting that transcriptional regulation of target genes by CARM1 through histone methylation might be largely associated with ACTR expression (Fig. 5c). More strikingly, ACTR knockdown by siRNA attenuated the mRNA expression of *SERPINE1* (Fig. 5d), implying that the coactivator CARM1 cooperates synergistically with ACTR, one of p160-type coactivators, to induce the expression of *SERPINE1*. Next, we investigated whether CARM1 recruitment at the *SERPINE1* gene requires ACTR, we conducted ChIP assay using

control MKN1 cells and ACTR knockdown MKN1 cells. As expected, reduced expression of ACTR markedly decreased the recruitment of CARM1 to the *SERPINE1* promoter (Fig. 5e). The luciferase activity of *SERPINE1* promoter was also decreased in ACTR-knockdown MKN1 cells compared to control MKN1 cells (Fig. 5f). We further examined whether silencing ACTR affected the migration and invasion of gastric cancer cells using Transwell migration and Matrigel invasion assays. Knockdown of ACTR dramatically decreased the migration and invasion of MKN1 cells (Fig. 5g).

Furthermore, overexpression of CARM1 alone did not significantly increase *SERPINE1* expression in MKN1 control cells, whereas simultaneous overexpression of CARM1 and ACTR increased the transcription of *SERPINE1* (Fig. 5h). However, this increase was not significant in exon 7-deleted MKN1 cells. On the other hand, CARM1 overexpression rescued *SERPINE1* expression in exon 7-deleted MKN1 cells. Our data suggest that LRRFIP2-variant 2 generated by exon7 deletion inhibits recruitment of CARM1 in combination with ACTR, one of p160 family of coactivators onto the *SERPINE1* promoter, leading to decreased expression of *SERPINE1*. Taken together, our results indicate that LRRFIP2 variant 2 suppresses the metastatic phenotype of gastric cancer by inhibiting the oncogenic function of CARM1.

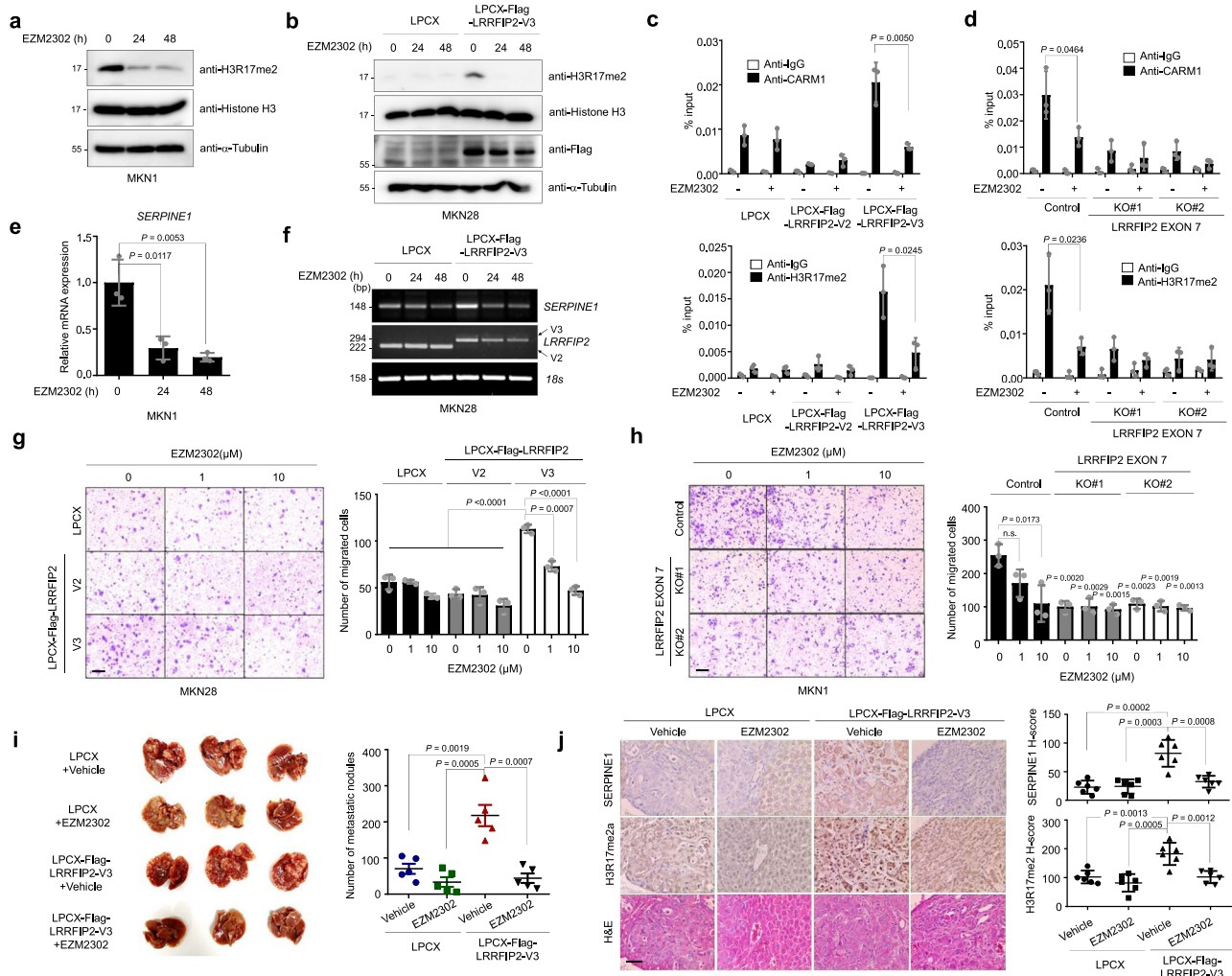

**Fig. 6 | EZM2302 reduces metastatic potential of gastric cancer cells by attenuating *SERPINE1* expression in the presence of LRRFIP2 variant 3.**
**a** Immunoblot analysis showing expression of H3R17me2a and histone H3 in MKN1 cells upon EZM2302 treatment for indicated hours. **b** Immunoblot analysis showing expression of H3R17me2a, histone H3, and Flag in MKN28 cells over-expressing LRRFIP2 variant 3 upon EZM2302 treatment for indicated hours.
**c, d** qChIP analysis of the human *SERPINE1* promoter upon treatment of EZM2302 for 24 h with antibodies to CARM1, H3R17me2a in **c** LRRFIP2 variant 2 and 3-overexpressing MKN28 cell lines, **d** LRRFIP2 variant 3 exon7 deleted MKN1 cell lines. **e** Real-time qRT-PCR analysis showing *SERPINE1* expression in MKN1 cells upon EZM2302 treatment for indicated hours. **f** RT-PCR analysis showing *SERPINE1* expression in MKN28 cells overexpressing LRRFIP2 variant 3 upon EZM2302 treatment for indicated hours. **g, h** Transwell migration assay and Matrigel invasion assay of **g** MKN28 cells overexpressing LRRFIP2 variant 2 and 3 and **h** LRRFIP2 variant 3 exon 7 deleted MKN1 cells with EZM2302 treatment (left) and bar graphs

showing number of invaded and migrated cells (right), respectively, following staining with crystal violet. Original magnification, _40X. Scale bar, 0.5 mm. *P* values verses Control 0 μM, otherwise indicated. **i** Representative whole liver image showing metastatic nodules (left) and scatter plot showing the number of liver metastatic nodules (right). **j** Representative IHC images showing SERPINE1 and H3R17me2a expression and H&E staining in metastasized liver tissues from **i** (left) and H-score scatter plots of the IHC staining (right). Original magnification, _200X. Scale bar, 50 μm. **a**, **b** The representative results were obtained from at least three independent experiments. **c**, **d** Data are representative mean ± SD of three independent samples (*N* = 3). **g**, **h** Data are representative mean ± SD of three independent experiments (*N* = 3). **i** Data are representative mean ± SD of five independent animals (*n* = 5). **j** Data are representative mean ± SD of six independent measurements (*n* = 6). All *P* values were calculated by unpaired two-tailed Student's t tests. n.s: not significant. Source data are provided in the Source Data file.

## Inhibition of CARM1 enzymatic activity represses *SERPINE1* expression and the invasiveness of LRRFIP2 variant 3-overexpressing gastric cancer cells

Given that CARM1 is an arginine methyltransferase, it would be of interest to determine whether the methyltransferase activity of CARM1 is critical for the transcriptional regulation of *SERPINE1* in gastric cancer cells. Treatment with EZM2302, a potent and selective inhibitor of CARM1 enzymatic activity[40], inhibited the enzymatic activity of CARM1, as shown by the markedly reduced asymmetric methylation of histone H3R17 (Fig. 6a). Furthermore, elevated methylation of histone H3R17 by ectopic overexpression of LRRFIP2 variant 3 was reduced by EZM2302 treatment, indicating that the induction of histone methylation in LRRFIP2 variant 3-overexpressing cells was indeed directed by

CARM1 (Fig. 6b). ChIP assays were performed on MKN28 cells over-expressing LRRFIP2 variants 2 and 3 and on MKN1 cells in which exon 7 was deleted following EZM2302 treatment with antibodies against CARM1 and asymmetric dimethylation of histone H3R17. Notably, inhibition of the enzymatic activity of CARM1 reduced both the recruitment of CARM1 and the enrichment of the asymmetric methylation of histone H3R17 on the *SERPINE1* promoter, which was enhanced by overexpression of LRRFIP2 variant 3 (Fig. 6c, d). Consequently, the mRNA expression of *SERPINE1* was reduced by treatment with EZM2302 (Fig. 6e). Upregulated *SERPINE1* expression by LRRFIP2 variant 3 was again downregulated by inhibitor treatment, supporting our hypothesis of the role of CARM1 as a transcriptional regulator of *SERPINE1* (Fig. 6f).

We next examined the migratory abilities of MKN28 and MKN1 cell lines upon treatment with EZM2302. Intriguingly, pharmacological inhibition of CARM1 dose-dependently abrogated the migratory advantage of LRRFIP2 variant 3-overexpressing cells over the control and variant 2-overexpressing MKN28 cells (Fig. 6g). Likewise, the migration of MKN1 control cells was dose-dependently reduced upon treatment, while no further reduction was observed in LRRFIP2 exon 7-deleted cells (Fig. 6h). We also investigated phenotypic changes in vivo by intrasplenic injection of LRRFIP2 variant 3-overexpressing cells into immunodeficient mice followed by EZM2302 treatment and observed a significant reduction in liver metastasis only in variant 3-overexpressing conditions (Fig. 6i, j). These results suggest that the activity EZM2302 in inhibiting metastasis of gastric cancer may depend on the degree of LRRFIP2 variant 3-expression. Interestingly, EZM2302 did not have any noticeable effect on either cell proliferation or foci-forming ability in these cell lines (Supplementary Fig. 20).

Collectively, these results consistently indicate that the inhibition of CARM1 enzymatic activity results in the repression of *SERPINE1* expression and subsequently abrogates the invasive potential of gastric cancer cells.

## Discussion

Although accumulating evidence suggests the downregulation of ESRP1 during the EMT process and the significance of its role in metastasis of many cancer types, the molecular mechanisms underlying its differential expression in each type of human cancer remain unclear. In this study, we discovered changes in the relative frequencies of distinct alternative splicing isoforms in gastric cancer cells significantly associated with the expression level of *ESRP1*. We found the two isoforms of *LRRFIP2*, whose expressions heavily depended on the expression of *ESRP1* and determined the metastatic fate of gastric cancer cells through differential protein-protein interaction with a methyltransferase protein, CARM1.

Studies have revealed that ESRP1 regulates epithelial- and mesenchymal-specific isoforms that have important roles in EMTs and disease processes such as cancer metastasis. RNA sequencing in a set of gastric cancer cell lines and gastric cancer patient tissues and iso-form switch analysis using iso-KTSP[15] provided a means to begin to interrogate ESRP1-induced alternative splicing events. Here, we focused on the splicing of *LRRFIP2*, which was the most significantly correlated with the expression of ESRP1. Until now, little is known about the function of LRRFIP2. It was first identified as a binding partner of Flightless-I and was found to modulate Wnt signaling through interactions with Dvl in Xenopus embryos and to inhibit NLRP3 inflammasome activation by recruiting the caspase-1 inhibitor Flightless-I[41–43]. However, its roles in cancer progression and metastasis have not been studied, perhaps due to the difficulties involved in verifying the distinct or perhaps opposite functions of its variants. Although there is a study in myocyte differentiation about distinct roles, expression profiles, and oligomerization properties of alternatively spliced isoforms of LRRFIP1, which exhibits 41% sequence homology with LRRFIP2, it did not determine which splicing factor or circumstance is responsible for the alternative splicing[44]. In this study, based on transcript profiles and isoform switch analysis of 18 gastric cancer cells and 18 gastric patient tissues, we demonstrated the significant association between the expression of ESRP1 and the alternative splicing of LRRFIP2 in gastric cancer cells.

In addition to *LRRFIP2*, other splicing candidates of ESRP1 in gastric cancer cells have been discovered by our RNA sequencing analysis. Among those genes, there were *CD44* and *FGFR2*, which are known splicing targets of ESRP1[2,45,46]. Although there are few studies about the roles of the splicing variants of these genes in gastric cancer[47,48], the correlations between the expression levels of ESRP1 and the relative frequencies of these variants in a set of gastric cancer cells and tissues have not been thoroughly examined until now. There were

also unknown targets of ESRP1, such as *CCDC50*, *BICD2*, and *CD47*, whose functional study would be noteworthy to extend our understanding about ESRP1-regulated splicing. Interestingly, the two splicing variants of *CCDC50* were recently found to be modulated by serine/arginine-rich splicing factor 3 (SRSF3), which determines tumor progression of hepatocellular carcinoma[49]. Considering that SRSF3 promotes tumor growth and metastasis while ESRP1 often suppresses them in various types of cancers, it would be an important finding to represent how alternative splicing of one single gene is regulated by two distinct splicing factors that give rise to two opposite phenotypes. This further study would allow us to better understand the mechanism underlying alternative splicing networks in various cell types and the cell- or tissue-specific splicing factors.

Recent studies have emphasized that the significant impact of alternative splicing is to remodel protein-protein interactions[50,51]. The epithelial and mesenchymal isoforms of Arhgef11, another splicing target of ESRP1 recently identified, provided a further example of how alterations in protein-protein interactions due to alternative splicing exhibit disease phenotypes[52]. Through mass spectrometry analysis, CARM1 was identified as a candidate binding partner exclusive to LRRFIP2 variant 2. Although minimal interaction between CARM1 and LRRFIP2 variant 3 was also observed, which could be possibly too weak to be detected in the mass spectrometry analysis, we confirmed much stronger interaction between CARM1 and LRRFIP2 variant 2. Our observation also supports the notion that alternative splicing alters protein-protein interactions and downstream effects. Additional studies on the identification of an independent binding partner of LRRFIP2 variant 3 would be required for deeper insight into the roles of LRRFIP2 variants.

Moreover, we discovered that altered transcription of *SERPINE1* in the exon 7 depleted cells was due to attenuated recruitment of CARM1 to the *SERPINE1* promoter and reduced asymmetric methylation of histone H3R17. Unlike other methyl transferases such as EZH2 and PRMT5, known to catalyze methylation for gene repression[53,54], CARM1 was found to promote gene activation by catalyzing asymmetrical dimethylation of R2, R17, R26 of histone H3[38]. Although we only examined the role of histone H3R17 in this study, we clearly observed the presence of CARM1 at the *SERPINE1* promoter coinciding with the increased level of histone H3R17 dimethylation. Consistent with the known functions of this methyltransferase and our results, we found decreased enrichment of CARM1 at the *SERPINE1* promoter and *SERPINE1* mRNA levels in CARM1 siRNA-transfected cells. Moreover, the transcriptional repression of *SERPINE1* was observed when the cells were treated with EZM2302, a potent CARM1 inhibitor, further supporting our hypothesis.

Our results additionally propose that deletion of exon 7 coinciding with the reduced methyltransferase activity of CARM1 may be due to their interaction competing with the interaction between CARM1 and ACTR. This finding was consistently supported by decreased histone H3R17 methylation at the *SERPINE1* promoter and mRNA expression level of *SERPINE1* in the absence of ACTR. Thus, we propose that the collaborative effect of ACTR and CARM1, which was competitively inhibited by LRRFIP2 in the absence of exon 7, could be responsible for these chromatin modifications for transcriptional regulation. Since ACTR is known to be often overexpressed in many cancer cells and primary tumors, including breast, ovarian, gastric, and prostate cancers[55–58], our findings show the possibility that tumor promoting activity of ACTR is due to the expression level of LRRFIP2 variant 3.

In conclusion, we identified the splicing variants of *LRRFIP2* whose expression levels were tightly regulated by ESRP1. *LRRFIP2* variant 2 which was dominantly expressed in *ESRP1*-high cells, inhibited the oncogenic function of CARM1 through its interaction, whereas variant 3 failed to inhibit the enzymatic activity of CARM1 in *ESRP1*-low cells. These results suggest that ESRP1 regulates the epithelial cell type-

specific splicing of LRRFIP2, thereby suppressing the metastatic potential of gastric cancer cells. In addition, although detailed studies and further validation are needed to assess their clinical significance, we propose that LRRFIP2 variants (2 or 3) may serve as potential biomarkers for gastric cancer liver metastasis and as therapeutic biomarkers for CARM1 inhibitors.

## Methods

### Cell Culture and Transfection

Total 18 gastric cancer cell lines (SNU-5, SNU-16, SNU-216, SNU-484, SNU-601, SNU-620, SNU-668, SNU-719, MKN-1, MKN-28, MKN-45, MKN-74, KATOIII, AGS, NCI-N87, SNU-1, SNU-520, and SNU-638), were purchased from the Korean Cell Line Bank. The cells were maintained under standard conditions (RPMI-1640 containing 25 mM HEPES, 10% fetal bovine serum, 100 unit/mL streptomycin, and 100 units/mL penicillin at 37 °C, 5% CO2). The 293 T cell line was cultured at 37 °C in DMEM medium with 10% fetal bovine serum (FBS) and 1% penicillin/streptomycin. Medium and reagents for cell culture were purchased from WELGENE, Inc., Republic of Korea. Transfections were carried out using Fugene® HD (Promega) according to the manufacturer's instructions.

### Plasmids

Human *LRRFIP2* variant 2 and 3 complementary DNAs (cDNAs) were amplified from MKN28 and MKN1 cDNAs, respectively, by PCR and subcloned into the XhoI and NheI sites of the pCS4-3Flag vector (Addgene). The two variants of Flag-LRRFIP2 was subcloned into the XhoI and HpaI sites of the LPCX vector (NIH), resulting in LPCX-Flag-LRRFIP2 variant 2 and variant 3.

### Generation of stable cell lines

For generation of retroviruses, GP2-293 cells were plated on a 100-mm culture plate 24 h before transfection. Transfection was performed using polyethylenimine (PEI) with 10 μg DNA and 5 μg VSV-G per plate. After transfection, the conditioned medium containing recombinant retroviruses was collected and filtered through 0.45-μm sterilization filters. Then, 3 ml of filtered retroviruses was applied immediately to MKN28 cells, which had been plated for 18 h before infection in a 100-mm culture dish. Polybrene (Sigma-Aldrich) was added to a final concentration of 8 μg ml−1, and the supernatants were incubated with the cells for 8 h. The medium was aspirated and replaced with fresh viral supernatant, and the procedure was repeated. After infection, the cells were placed in fresh growth medium for 24 h and cultured as usual. Selection with 2 μg ml−1 puromycin (Sigma-Aldrich) was initiated 48 h after infection.

### CRISPR genome editing

To delete exon 7 of LRRFIP2 variant 3, A guide RNA (gRNA) targeting exon7 of LRRFIP2 variant3 was designed (gRNA1: 5′-CCTCCATATA-TAGCCC TGTCCCC-3′; gRNA2: 5′-CCGTGGTGTCTTAGCCATACAAA-3′). Oligonucleotides were synthesized and ligated into pSpCas9(BB)−2A-Puro (PX459) as previously reported[59]. MKN1 cells were transfected with the vectors containing gRNA sequence using the Neon Transfection System (Thermo Fisher Scientific) following the manufacturer's protocol. Then we went through clonal selection of the cells and checked the expression by RT-PCR.

### RNA extraction, RT-PCR and real-time qRT-PCR

Total RNA was isolated from cells and tissues using the easy-BLUE Total RNA extraction kit (Intron bio) according to the protocol provided by the manufacturer. Reverse transcription was carried out with 2 μg of purified RNA using M-MLV reverse transcriptase (Promega, M1705). The synthesized cDNA was amplified by PCR using specific primers. PCR products were visualized by electrophoresis on 1.5% −2% agarose gels with Redsafe (Chembio, 21141) staining and analyzed with

an ImageQuant LAS 4000 image analyzer (GE Healthcare). The 18 S rRNA gene was used as an internal control.

A forward primer of 5′-CCTCAGCAACAACCCCTCTA-3′ and a reverse primer of 5′-CCTGCTCTTCAATAACATCC-3′ were used to detect LRRFIP2 variants 2 and 3. The PCR products of LRRFIP2 variant 2 and 3 are 222 bp and 294 bp, respectively. Quantitative real-time PCR (qRT-PCR) was performed with the proper primers using 2× SYBR Green PCR Master Mix (TaKaRa) and conducted by QuantStudio 5 (Applied Biosystems). All of the primers used in this study is listed in Supplementary Table 1.

### RNA-seq data analysis of gastric cancer cell lines and tissues

We reanalyzed our previously published RNA-seq data obtained from the 18 gastric cancer cell lines, 18 gastric cancer tissue samples, and 16 normal gastric tissue samples[14]. For heatmaps, we subtracted Pearson's correlation coefficient of the expression of ESRP1 and the 2nd set of isoforms from Pearson's correlation coefficient of the expression of ESRP1 and the 1st set of isoforms in the tissue data. Then we neglected the genes whose sum of the average of the relative expression of isoform 1 and isoform 2 is too small (less than 0.5). The top 20 genes in the resulting list were used to generate the heatmaps in Figs. 1b, c and 2b, c. In order to quantify transcript-level abundances without aligning the RNA-seq reads, Salmon version 0.8.0[60] was used with Refseq annotation for GRCh38 genome. The quantified transcript levels were normalized by TPM (Transcript Per Million). We obtained the top 100 transcript variants differentially expressed by ESRP1 expression level using a isoform identification software iso-kTSP[15]. Using SUPPA[17], we generated the seven splicing event types Alternative 3′ Splice Sites (A3), Alternative 5′ Splice Sites (A5), Alternative First Exon (AF), Alternative Last Exon (AL), Mutually Exclusive Exon (MX), Retained Intron (RI), and Skipping Exon (SE) by ESRP1 in gastric cancer cell lines. We confirmed the ESRP1 expression in large-scale gastric cancer tissue downloaded from TCGA database[61]. The expression of LRRFIP2 isoforms was obtained from SpliceSeq[62].

### RNA sequencing of LRRFIP2 (Δexon7) cell lines

An Illumina platform (Illumina) was used to analyze transcriptomes with a 151 bp paired-end library. The cDNA libraries were prepared from Total RNA for 151 bp paired-end sequencing using TruSeq stranded mRNA Sample Preparation Kit (Illumina). Total RNA from each cell for RNA sequencing was isolated using TRIzol reagent following the manufacturer's instructions. The mRNA molecules were purified and fragmented from 1ug of Total RNA using oligo (dT) magnetic beads. After sequential process, cDNA libraries were amplified with PCR (Polymerase Chain Reaction) and were subsequently examined for quality using an Agilent 2100 bioanalyzer (Agilent). They were quantified with the KAPA library quantification kit (Kapa Biosystems) according to the manufacturer's protocol. Following cluster amplification of denatured templates, samples were pair-end sequenced with the Illumina Novaseq6000 platform (Illumina). After sequencing, low quality reads were filtered out according to the following criteria by using cutadapt v.2.8[63]: The adapter sequences and the ends of the reads less than Phred quality score 20, and simultaneously the reads shorter than 50 bp. The whole filtering process was performed using scripts developed in-house. Filtered reads were mapped to the GRCh38 genome related to the species using STAR v.2.7.1a alignment software[64]. Gene expression levels were measured with RSEM v1.3.1[65] using the ensemble database and quantified as the ratio of reads mapped to a gene to the gene length in kilobases and expressed as the fragments per kilobase of transcript per million fragments mapped (FPKM). Noncoding gene regions were excluded from gene expression measurement. To improve the accuracy of the measurements, the multiread correction and frag-bias-correct options were applied. All other options were set to their default values.

**GO, KEGG pathway, and PPI network analysis.** The enriched GO and KEGG pathway terms for Figs. 1d, e and Supplementary Fig. 3d, e were obtained from Enrichr software[66]. DAVID tool[67] and the KEGG orthology-based annotation system (KOBAS) online tool[68] were used for Fig. 3b with cut-off values of $P < 0.01$. String was used to generate protein-protein interaction (PPI) networks[69].

**Sashimi plots.** To generate sashimi plot for the splicing variants LRRFIP2, CCDC50 and BICD2, we aligned the RNA-seq reads on the GRCh38 genome using STAR[64]. Next, we combined the mapped files into four files ESRP1-low cell lines, ESRP1-high cell lines, ESRP1-low tumor tissues (137 T, 87 T, 236 T, 211 T, 80 T, 135 T, and 134 T), and ESRP-high tumor tissues (130 T, 134 T, 103 T, 95 T, 195 T, 849 T, 43 T, 917 T, 859 T, 119 T, 889 T, and 882 T) according to the reference[70]. We drew the Sashimi plot of the genomic locus of LRRFIP2, CCDC50 and BICD2 using Integrative Genomics Viewer (IGV)[71].

**Analysis of patient survival.** Patients enrolled in this study had measurable and histologically or cytologically confirmed metastatic and/or recurrent gastric adenocarcinoma[19]. The trial was conducted in accordance with the Declaration of Helsinki and the Guidelines for Good Clinical Practice (ClinicalTrial.gov identifier: NCT# 02628951). The trial protocol was approved by the Institutional Review Board of Samsung Medical Center (Seoul, Korea), and all patients provided written informed consent before the enrolment. Kaplan-Meier survival analysis was performed using gene expression profiles and clinical data from stage IV gastric cancer patients ($N = 37$). For survival analysis, R package 'survival' and 'survminer' were used in R version 3.5.3.

**Immunoprecipitation and immunoblot analysis**
Cells were washed twice in cold PBS and lysed in IP buffer (50 mM Tris, pH 7.4, 150 mM NaCl, 1% Triton X-100, 0.5% sodium deoxycholate, 2 mM EDTA and 10% glycerol) plus phosphatase and protease inhibitors (Roche). Whole-cell extracts were incubated with the appropriate primary antibodies overnight at 4 °C. Antibody-bound proteins were precipitated with Dynabeads (Thermo Fisher Scientific) to the manufacturer's protocol. The beads were washed three times with lysis buffer and then eluted in 2× SDS sample loading buffer. Eluted proteins were separated by SDS–polyacrylamide gel electrophoresis, transferred to PVDF membranes (Millipore), and detected using appropriate primary antibodies coupled with a horseradish peroxidase-conjugated secondary antibody by chemiluminescence (GE Healthcare). All of the antibodies used in this study are listed in Supplementary Tables 2 and 3.

**RNA immunoprecipitation**
RNA immunoprecipitation was performed as previously described[72]. The total cell protein extracts were obtained by incubating MKN28 cells for 5 min in cold isotonic buffer (20 mM HEPES, 100 mM NaCl, 250 mM Sucrose, 5 mM MgCl2), a cocktail of protease inhibitors (Roche) and RNAse inhibitor (Promega) and DTT. The lysates were precleared for 1 h at 4 °C using Dynabeads protein G. Anti-ESRP1 antibody (Sigma-Aldrich) or rabbit IgG was added to the precleared lysates overnight at 4 °C and the day after, dynabeads were added for 1 h at 4 °C.

**Crosslinking for protein interaction analysis**
Crosslinking of LRRFIP2 variant 2 and 3-overexpressing MKN28 cells for mass spectrometry analysis was performed as described previously[73]. In particular, cells were fixed in 1% formaldehyde in PBS. Eluted proteins were loaded on SDS-polyacrylamide gel and separated by electrophoresis. Crosslinked proteins on gel were stained in Coomassie blue (Tech & Innovation) and analyzed through mass spectrometry at National Instrumentation Center for Environmental Management (NICEM) (Seoul, Korea).

**Cell migration and invasion assays**
Transwell migration assays were performed using Transparent PET membrane inserts (Falcon, 353097) as described in the manufacturer's protocol. A total of $1 \times 10^5$ cells were plated in the insert and incubated for 16 h. Invasion assays were performed with BioCoat Matrigel invasion chambers (Corning, 354578) as described in the manufacturer's protocol. Cells were starved in DMEM medium without FBS for 24 h. Starved cells ($1 \times 10^5$) were plated in the top chamber, which contained serum-free DMEM, and the bottom chamber contained DMEM with 10% FBS. After 24 h of incubation, noninvasive cells were removed with a cotton swab. The cells that migrated through the membrane and adhered to the lower surface of the membrane were fixed with 70% ethanol and stained with 0.05% crystal violet. The numbers of invaded cells in each field of view were quantified for statistical analysis.

**In vivo tumor formation and liver metastasis.** All experimental protocols were approved by the Institutional Animal Care and Use Committee of Center at Woojung Bio, (Suwon, Korea). The laboratory mice were maintained on a 12-h light/dark cycle at room temperature (20–22 C) with constant humidity ($40 \pm 10\%$). For the tumor-formation assay, a total of $1 \times 10^7$ retrovirus-infected MKN28 cells were resuspended in 1:3 PBS/hydrogel (The Well Bioscience) solution and subcutaneously injected into 6-week-old male NOD/ShiLtJ-Prkdc[emlAMC]Il2rg[emlAMC] (NSGA, Joong Ah Bio) mice ($n = 5$ per group) to measure tumor growth. We did not allow the tumor size to exceed 20 mm in diameter as it is the maximal tumor size permitted by the ethics committee. In liver metastatic model, the skin was shaved and rubbed with ethanol pads, and a 0.5 cm abdominal incision was made adjacent to the spleen. $1.5 \times 10^6$ cells were suspended in 100 μl of PBS and injected into the spleen of 6-week-old male NOD/ShiLtJ-Prkdc[emlAMC]Il2rg[emlAMC] (NSGA, Joong Ah Bio) mice ($n = 4$ or 5 per group) under general anesthesia. The needle was maintained in the spleen tissue for two minutes following injection. A surgical suture was placed across the hilum of the spleen to prevent bleeding, and a splenectomy was then performed. Five weeks after the injection, the mice were sacrificed and the livers were removed and prepared for histological examination (hematoxylin and eosin (H&E) and IHC (Immunohistochemistry) staining). For inhibitor study, EZM2302 or vehicle (0.5% methylcellulose in dH20) was administered orally BID at a dose of 100 mg/kg for 21 days.

**Quantification and statistical analyses**
The $P$ values in Figs. 1o, 3f, and Supplementary Figs. 1, 12d, and 13b were calculated using log-rank test. The $P$ values in Figs. 1k-n, 3e, and Supplementary Fig. 12c were calculated by two-sided Wilcoxon rank sum tests. For all other comparisons, the two-tailed unpaired Student's t-test in GraphPad Prism 5 was used, and $P < 0.05$ indicated statistical significance.

**Reporting summary**
Further information on research design is available in the Nature Research Reporting Summary linked to this article.

## Data availability
We reanalyzed our previously published RNA-seq data which is available in the NCBI Sequence Read Archive (SRA) (http://www.ncbi.nlm.nih.gov/sra) under accession number SRP014574. RNA sequencing data for Fig. 3a has been deposited in the NCBI GEO under accession code GSE194309. Relapse-free survival graphs for Fig. 3f and Supplementary Figs. 1, 12d and 13b were analyzed by the Kaplan−Meier Plotter analysis tool (http://kmplot.com/analysis). The source data for Fig. 1a, b, c, d, e, h, j, o, 2a, e, f, k, l, 3a-d, 4h, i, j, k, 5d, e, f, g, h 6c, d, g, h, i, j and Supplementary Figs. 3a–e, g 4a, b, 5b, 6, 8a, 10a, 11a, d, 12a, b, 16a, b, 17a, 19d, and 20a–e have been provided as Source Data file.

Unprocessed original scans of blots are shown in Source Data file. Source data are provided with this paper.

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

## Acknowledgements

We thank Satoru Takahashi for kindly providing us pSpCas9(BB)–2A-Puro (PX459) vector containing gRNAs targeting LRRFIP2. This research was supported by a grant of the Basic Research Support Project funded by GILO Foundation, Republic of Korea.

## Author contributions

S.J.K. conceived and supervised the project and wrote the manuscript. K.M.Y. designed the study, supervised the project, and made substantial contributions to the final manuscript. Jihee L. and K.P. designed experiments, collected the data, performed the analysis, and wrote the manuscript. J.K. conducted RNA sequencing and bioinformatics analyses. E.H., M.S. and S.P contributed to data collection. Jeeyun L. and H.C. collected and analyzed the survival data of patients. M.L. performed immunohistochemical analysis. H.K.Y contributed to gastric patient tissue analysis and interpretation. J.P., A.O., and K.S.P. contributed to data analysis and interpretation. All authors discussed the results and gave final approval of the version of the manuscript to be submitted.

## Competing interests

The authors declare the following competing interests: S.J.K. has personal financial interests as a shareholder in Medpacto Inc. All other authors declare no competing interests.
