## [Peer Review File · Nature Communications]

ESRP1-regulated isoform switching of LRRFIP2 determines metastasis of gastric cancerREVIEWER COMMENTS

Reviewer #1 (Remarks to the Author):

In this manuscript, the authors investigated how ESRP1 regulation of LRRFIP2 alternative splicing in gastric cancer can affect liver metastasis. The authors showed that ESRP1 expression in gastric cancer cells can determine the alternative splicing of a group of genes including the leucine-rich repeat Fli-I-interacting protein 2 (LRRFIP2). The authors showed that the LRRFIP2 v3 splice isoform, which is highly expressed in ESRP1-low gastric cancer cells, promotes liver metastasis. They further used RNA-seq and identified several downstream targets of LRRFIP2, including SERPINE1. They showed that LRRFIP2 v2 preferentially interacts with the arginine methyltransferase CARM1 to activate transcription of SERPINE1. The authors proposed a potential mechanism that V3 increases CARM1 binding to ACTR, which facilitates docking of CARM1 on the SERPINE1 promoter. Thus, the authors connected several proteins in the pathways to regulate gastric cancer, yet these observations are mostly at the level of correlations. As an example, one of the key points of this manuscript was ESRP1 promotes a non-metastatic LRRFIP2 isoform, inhibiting gastric cancer metastasis. There is no data presented to examine whether ESRP1 directly regulates splicing of LRRFIP2. There is also lack of functional/causal relationships examined between ESRP1 and LRRFIP2.

Main comments:

1. The authors seem to be misusing the concept of PSI (percent spliced in). PSI quantifies the percentage of inclusion of a specific exon, and it is not a direct representation of the transcript being expressed. The PSI of LRRFIP2 V2 would be 0% while the PSI of V3 would be 100%. The authors should relabel all the V2 PSI and V3 PSI as the percentage expression of V2 or V3. The authors should re-evaluate all the figures that used PSI for each transcript and replot the figures in the correct way, including Fig 1h-i, Fig 2g, Supp Figs 4, 5, and 7.
2. Some of the bioinformatics data and splicing analysis should be revised to reflect the findings more accurately. The heat maps of Fig. 1b and 1c are incorrect. Unless the authors describe different splice isoforms in ESRP1-low and ESRP1-high, only the heat map in 1b should be shown. This also applies to Fig. 2b, 2c. Moreover, the authors should use sashimi plots to visualize RNA-seq reads that indicate the difference in isoform usage (Fig 1g, Supp Fig 3).
3. The authors found the LRRFIP2 splicing phenotype through analysis of ESRP1 expression, but did not address whether ESRP1 directly promotes LRRFIP2 exon 7 skipping. As ESRP1 regulation of LRRFIP2 is a major conclusion of this manuscript, the authors should determine whether ESRP1 directly promotes LRRFIP2 exon 7 skipping or whether this is an indirect effect.
4. The authors performed ESRP1-dependent GO and KEGG analyses in gastric cancer cell lines (Fig. 1d,1e) and in patient specimens (Fig. 2d, 2e). The authors should reconcile the results and document whether the patient dataset recapitulate the cell line findings, in contrast to current separated figures and conclusions.
5. Fig. 3d, levels of LRRFIP2 isoforms need to be shown using the LRRFIP2 antibody and not the flag-tagged antibody to show both endogenous and ectopically expressed LRRFIP2 isoforms. At least the isoform levels should be shown by RT-PCR.
6. In Fig 5, the authors identified CARM1 as a LRRFIP2 interacting partner and showed that V2 has higher affinity binding to CARM1. The authors then discussed how V3 expression can enhance CARM1 methylation activity on the SERPINE1 promoter. LRRFIP2 is located in cytoplasm, whereas CARM1 is in the nucleus. What is the functional connection between LRRFIP2 variants and CARM1?
7. The authors stated that CARM1 directly regulates SERPINE1 transcription and showed that CARM1 binds the SERPINE1 promoter by ChIP. The authors are required to directly examine the transcription activity of CARM1 on the SERPINE1 promoter, for example, using promoter luciferase assays. It is necessary to experimentally demonstrate 1) CARM1 and ACTR directly regulate SERPINE1 transcription; 2) CARM1 and ACTR interaction is differentially affected by LRRFIP2 isoforms; and 3) CARM1 and ACTR are the direct downstream effectors of LRRFIP2 that regulate SERPINE1 transcription.

Minor points:

1. Some of the invasion/migration assays and the western blots showed subtle differences. The authors should optimize experimental conditions before further conclusions can be drawn.

- Fig 5i, Fig 6a, f: the differences are subtle
- Fig 5d (CARM1 KD), Fig 6c (siGRIP1, siSRC1): the KD efficiency seems very poor, or the antibodies are not good
- Fig 6b: the quality of the blots is not clear enough to tell whether any of the conclusions are reliable

2. Several figures have p values and significance levels that are questionable, and a few figures have mistakes that needed proofreading:

- Fig 1d-e, Fig 2d-e, Fig 4b: the p value of the GO and KEGG pathways seem too big for analyzing a standard differentially expressed gene list.
- Fig 2m, Fig 5i: figures need proofreading
- Fig 2i-m, Fig 4e: the difference in TCGA subgroups is very subtle. The authors also did not explain why the sample numbers of all the box-and-whisker plots are different.
- Fig 7a-b: would suggest rerunning the samples

3. The authors have been using the exon 7 KO cell lines as a loss-of-function of LRRFIP2 V3. However, from Fig 3g, the KO is also a gain-of-function of V2. The authors should at least comment on the possibility that the exon 7 KO phenotypes could also be caused by the gain of V2 in addition to the loss of V3.

4. "an exon 7-truncated form" should be stated as exon-7 skipped form

5. 1st paragraph 2nd line – "As in the previous study about ESRP1..." – cite the reference.

6. Some of the Supplemental figures should be combined to pair with the main figures.

In conclusion, the authors showed strong evidence that the LRRFIP2 V3 promotes gastric cancer liver metastasis. Some of the data in this manuscript needs more refinement. Importantly, the authors are required to provide causal evidence to support the axis of ESRP1 – LRRFIP2 splicing – CARM1 methylation – SERPINE1 transcriptional activation – metastatic phenotypes.

Reviewer #2 (Remarks to the Author):

The manuscript by Lee et al entitled "ESRP1-regulated isoform switching of LRRFIP2 determines metastasis of gastric cancer" reported the identification and functional characterization of a novel ESRP1 RNA splicing target gene LRRFIP2 in gastric cancer. Mechanistically, the authors propose that variant 2 of LRRFIP2 competes with ACTR for CARM1 binding to suppress CARM1 activity to inhibit metastasis. The functional data of the manuscript is interesting with translational implications. However, the major weaknesses of the manuscript will have to do the mechanism of action and scientific rigor. The authors should consider the following comments to support their conclusions.

General comments

1. Mechanism of action: The notion for fig. 6 that LRRFIP2-v2 interacts with CARM1 to compete with ACTR is not convincing. The interaction between CARM1 and ACTR in fig. 6b was not decreased by overexpression of v2. And this notion cannot explain why overexpression of v3 increased the interaction between CARM1 and ACTR. The notion also cannot explain why overexpression of v2 did not affect invasion and migration while overexpression of v3 promoted invasion and migration. Instead, all the results in fig. 6 suggested that v3 promotes the interaction between CARM1 and ACTR and v2 variant has no effect on the interaction. This is consistent with fig. 6a that conversion of v3 to v2 by CRISPR KO of exon 7 decreased the expression of v3, and therefore resulted in a decreased interaction between CARM1 and ACTR. However, this still cannot explain Fig. 5b that v2 dominantly interacts with CARM1. Overall, the proposed mechanism of action cannot explain the phenotypes reported in this study. In addition, perhaps more

importantly, the authors need to explore how v3 promotes CARM1 activity (and possibly the interaction between CARM1 and ACTR) to mediate the metastasis-promoting effects of LRRFIP2 v3.

2. Along these lines, genome-wide unbiased approaches such as ChIP-seq and RNA-seq could be helpful in addressing the missing mechanistic links among CARM1, LRRFIP2 v2 vs. v3, and ACTR.
3. The ChIP analysis (should use qPCR analysis) and RT-qPCR analysis should be quantified with at least three biological repeats (agarose gel imaging is not sufficient in this day and age).
4. For functional studies (at least for in vitro), please include more than one cell line to limit potential cell line specific effects.

Specific comments:

1. Fig. 2a, please specify how the cut-off was determined for Fig. 2b and 2c.
2. Fig. 2i-l. The correlation analysis between ESRP1 expression and LRRFIP2v3 in TCGA dataset is not proper. The numbers of TCGA subgroups are different between ESRP1 expression and LRRFIP2v3 PSI groups. Since they are in the same TCGA cancer type, the numbers of each subgroup should be the same. Correlation should be done in the same sample pool.
3. In fig. 2m, is ESRP1 expression positively correlated with survival in this dataset?
4. Fig. 4f and Supplementary Fig. 12b used auto select best cutoff function in Kaplan-Meier plotter for the survival plot, which is not proper. Median or quartile split of patients is more objective.
5. Fig. 5, please provide evidence of interaction between CARM1 and LRRFIP2 v2 and v3 at the endogenous level. Please quantify Fig. 5k by using H score. In addition, IHC against different targets such as H3R17me2a and SERPINE1 should be performed in serial sections and pictures should be taken in the roughly same regions for comparison. The pictures seem to be taken in random regions.
6. Fig. 5g, Overexpression of V3-LRRFIP2 should not affect the endogenous level of V2-LRRFIP2. Thus, a lower band of V2-LRRFIP2 should be observed.
7. The conclusion for Fig. 5 is not proper. There is no evidence suggest that LRRFIP2-v3 is critical for CARM1 enzymatic activity. Fig. 5 suggests that LRRFIP2-v3 affect the recruitment of CARM1 to target genes.
8. The data as presented in Fig. 6a do not appear to support the conclusion that exon 7 knockout only affects CARM1's interaction with ACTR (as both GRIP1 and SRC1 appear to show a similar decrease in interaction). Likewise, in Fig. 6c, GRIP1 and SRC1 knockdown also reduced expression of SERPINE1. In addition, 6a anti-GRIP1 and anti-SRC1 IP blots look similar. Anti-GRIP1 IP immunoblot in 6b was not successful. In 6a, SRC1 and GRIP1 were detected as a single band in WCL but two bands in IP blots and 6b immunoblots. Immunoblots for CARM1 are also not consistent with a single band in 6a and 6b IP while two bands in 6b WCL. Please clarify.
9. The description for Fig. 7c-d is not proper. CARM1 inhibitor EZM2302 did not affect the recruitment of CARM1 to SERPINE1 promoter.
10. What is the expression/amplification status of ESRP1 in gastric cancer in TCGA?
11. Scale bars and molecular weight markers are missing for all the figures.

Reviewer #3 (Remarks to the Author):

The study investigates the effect of different variants of LRRFIP2 in gastric cancer metastasis. The authors argue that LRRFIP2 variant 3 but not variant 2 induces metastasis via coactivating transcription of SERPINE1 with CARM1. The authors also argue that LRRFIP2 variant 3 is present mostly in tumors with low expression of ESRP1, as ESRP1 is responsible for processing LRRFIP2 variant 3 into variant 2, and low ESRP1 expression or high LRRFIP2 variant 3 correlates with poor patient survival. The study intriguingly argue that different variants might function differently, which has been often overlooked. However, a few major concerns remain for this investigation.

1. There is no analysis on the protein levels of the two LRRFIP2 variants. Only RT-PCR assessing the mRNA levels was shown. Since one exon is deleted in the variant 2, there should be a change of molecular weight which should be detectable in western blot. Also, for the RT-PCR measure mRNA levels of these variants, the authors should provide a schematic showing how the primers flank the regions of different exons; and the primer sequences should be provided in the Methods.
2. The CRISPR approach deleting exon 7 of LRRFIP2 is doubtful. Again, a western blot analysis is critical to make the judge whether only LRRFIP2 v3 is knockout leaving v2 intact. Assessing mRNA with RT-PCR is not a way to measure CRISPR knockout. The authors could also sequence the knockout clones and show the mutations generated by the knockout. CRISPR knockouts a gene by generating premature stop codon, so probably both v2 and v3 have been knockout. To target v3

only, the authors can consider using shRNAs specifically targeting exon7.

3. The authors should provide in vivo data for the drug targeting CARM1.

4. Cell line models are scarce for the functional part in this study, with just one for overexpression and one for KO. The authors should try to perform knockout or knockdown in two independent cell lines.

Apart from these major concerns, there are also a couple of minor ones.

1. The authors could consider providing metastasis free survival apart from overall survival as the investigation is mostly for metastasis.

2. The authors could start with tumor tissue data (fig 2) instead of cell line data (fig 1) showing LRRFIP2 and ESRP1 as cell line data becomes supportive when tumor tissue data is available.

ESRP1-regulated isoform switching of LRRFIP2 determines metastasis of gastric cancer

#NCOMMS-21-22414A

Response to Reviewers

We greatly appreciate the constructive comments from the reviewers and the invitation from the editor to submit a revised version. We have thoroughly revised the manuscript following the reviewers' suggestions. Please see below our point-to-point responses in non-italic text following reviewer comments in italic text.

REVIEWER #1

1. The authors seem to be misusing the concept of PSI (percent spliced in). PSI quantifies the percentage of inclusion of a specific exon, and it is not a direct representation of the transcript being expressed. The PSI of LRRFIP2 V2 would be 0% while the PSI of V3 would be 100%. The authors should relabel all the V2 PSI and V3 PSI as the percentage expression of V2 or V3. The authors should re-evaluate all the figures that used PSI for each transcript and replot the figures in the correct way, including Fig 1h-i, Fig 2g, Supp Figs 4, 5, and 7.

Response: As Reviewer #1 pointed out, we agree that the term PSI has been misused. We tried to show the relative TPM of each variant. We modified the term "PSI" to "Relative expression" for Fig 1h-I (i and k in revised Fig 1), Fig 2g (f in revised Fig 2), Supp Figs 4, 5, and 7 (S8 in revised fig Supp Figs).

2. Some of the bioinformatics data and splicing analysis should be revised to reflect the findings more accurately. The heat maps of Fig. 1b and 1c are incorrect. Unless the authors describe different splice isoforms in ESRP1-low and ESRP1-high, only the heat map in 1b should be shown. This also applies to Fig. 2b, 2c. Moreover, the authors should use sashimi plots to visualize RNA-seq reads that indicate the difference in isoform usage (Fig 1g, Supp Fig 3).

Response: We understand Reviewer #1's point. As we mentioned above, we misused the term PSI. We attempted to show the relative abundance of a set of isoforms highly expressed in *ESRP1*-low condition in Fig. 1b and Fig. 2b, and the relative abundance of another set of isoforms (distinct from the first set) highly expressed in *ESRP1*-high condition in Fig. 1c and Fig. 2c. Again, to avoid confusion, we have replaced the terms. RefSeq information for these isoforms is given in Supp Table SI.

As suggested by Reviewer #1, we aligned RNA-seq reads on the GRCh38 genome using STAR to generate sashimi plots for splicing variants of the LRRFIP2, CCDC50 and BICD2 genes, (Dobin et al., 2013). Next, we combined the mapped files into four files *ESRP1*-low

cell lines, *ESRP1*-high cell lines, *ESRP1*-low tumor tissues (137T, 87T, 236T, 211T, 80T, 135T, and 134 T), and *ESRP*-high tumor tissues (130T, 134T, 103T, 95T, 195T, 849T, 43T, 917T, 859T, 119T, 889T, and 882T) according to the reference (Li et al., 2009). Sashimi plots of the genomic locus of *LRRFIP2*, *CCDC50* and *BICD2* were drawn using Integrative Genomics Viewer (IGV) (Robinson et al., 2011). We have included these new results in Fig 1h, 2e, Supp Fig S3b and S3d.

Fig. A. Splicing variants of *LRRFIP2* differentially display exon skipping events in *ESRP1*-low and *ESRP1*-high conditions. Sashimi plots of the genomic locus of *LRRFIP2* in **a** gastric cancer cell lines and **b** gastric cancer patient tissues. The indicated exon numbers are the ones of *LRRFIP2* variant 3 (NM_001134369).

Fig. B. Splicing variants of *CCDC50* and *BICD2* differentially display exon skipping events in *ESRP1*-low and *ESRP1*-high conditions. Sashimi plots of the genomics locus of **a *CCDC50* and **b** *BICD2* shows splice junctions from the aligned RNA-seq data (Fig. 1b, c and 2b, c).**

References

- Dobin, A. *et al.* STAR: ultrafast universal RNA-seq aligner. *Bioinformatics* **29**, 15-21 (2013).
- Li, H. *et al.* The Sequence Alignment/Map format and SAMtools. *Bioinformatics* **25**, 2078-9 (2009).
- Robinson, J.T. *et al.* Integrative genomics viewer. *Nat Biotechnol* **29**, 24-6 (2011).

3. The authors found the *LRRFIP2* splicing phenotype through analysis of *ESRP1* expression, but did not address whether *ESRP1* directly promotes *LRRFIP2* exon 7 skipping. As *ESRP1* regulation of *LRRFIP2* is a major conclusion of this manuscript, the authors should determine whether *ESRP1* directly promotes *LRRFIP2* exon 7 skipping or whether this is an indirect effect.

Response: It has been demonstrated that *ESRP1* expression is strongly downregulated during the epithelial-mesenchymal transition (EMT) and splicing signatures are broadly associated with cancer cells with EMT features. Even though a large number of *ESRP1* target genes have been identified and many papers have been published on them, how *ESRP1* regulates splicing programs, including exon skipping, is still not well understood (Bhattacharya et al., 2018; Lee et al., 2018; Yae et al., 2012). It would be interesting to investigate how *ESRP1* promotes *LRRFIP2* exon 7 skipping, but this is beyond the scope of our current study. This question remains to be solved in the future study.

Among the genes identified as well-known targets of *ESRP1*, *CD44* and *FGFR2* were also identified in our study (Supplementary Table S1). Furthermore, we have also observed that most gastric cell lines in *ESRP1*-low group was found to be mesenchymal subtype (Supplementary Fig. 2). Based on these findings, we believe that *ESRP1* directly promotes *LRRFIP2* exon 7 skipping. To further support our finding, we also investigated *ESRP1*-mediated *LRRFIP2* alternative splicing switch in the SNU484 gastric cancer cell line in addition to MKN1. **Fig. C** shows that *ESRP1* expression induces exon skipping in both MKN1 and SNU484 cell lines. We have included these results in Fig. 3b.

Fig. C. Overexpression of *ESRP1* induces variant switch of *LRRFIP2* in MKN1 and SNU484 cells. RT-PCR analysis shows expression of *LRRFIP2* variants and *ESRP1* upon overexpression of *ESRP1*.

References

Bhattacharya, R., Mitra, T., Ray Chaudhuri, S. & Roy, S.S. Mesenchymal splice isoform of *CD44* (*CD44s*) promotes EMT/invasion and imparts stem-like properties to ovarian cancer

cells. *J Cell Biochem* **119**, 3373-3383 (2018).

Lee, S. *et al.* Esrp1-Regulated Splicing of Arhgef11 Isoforms Is Required for Epithelial Tight Junction Integrity. *Cell Rep* **25**, 2417-2430 e5 (2018).

Yae, T. *et al.* Alternative splicing of CD44 mRNA by ESRP1 enhances lung colonization of metastatic cancer cell. *Nat Commun* **3**, 883 (2012).

4. The authors performed ESRP1-dependent GO and KEGG analyses in gastric cancer cell lines (Fig. 1d,1e) and in patient specimens (Fig. 2d, 2e). The authors should reconcile the results and document whether the patient dataset recapitulate the cell line findings, in contrast to current separated figures and conclusions.

Response: It is widely appreciated that gastric cancer cells are characterized with extensive intertumoral and intratumoral heterogeneity (Cancer Genome Atlas Research, 2014; Wang et al., 2021; Wang et al., 2020). Despite of the heterogenic characteristic of gastric cancer tissues, we tried to demonstrate that the ESRP1-dependent GO and KEGG analyses in the patient specimens are relatively consistent with those in the cell lines in the aspect of cell junction, migration and proliferation. However, as Reviewer #1 has suggested, we reconciled the conclusion in the Result (p7, line 161-165) and rearranged the figures and put the GO and KEGG pathway analyses in Supplementary Fig. 7.

References

Cancer Genome Atlas Research, N. Comprehensive molecular characterization of gastric adenocarcinoma. *Nature* **513**, 202-9 (2014).

Wang, R. *et al.* Single-cell dissection of intratumoral heterogeneity and lineage diversity in metastatic gastric adenocarcinoma. *Nat Med* **27**, 141-151 (2021).

Wang, R. *et al.* Multiplex profiling of peritoneal metastases from gastric adenocarcinoma identified novel targets and molecular subtypes that predict treatment response. *Gut* **69**, 18-31 (2020).

5. Fig. 3d (Fig. 3e in revised Figure 3), levels of LRRFIP2 isoforms need to be shown using the LRRFIP2 antibody and not the flag-tagged antibody to show both endogenous and ectopically expressed LRRFIP2 isoforms. At least the isoform levels should be shown by RT-PCR.

Response: In response to Reviewer #1's comments, we have conducted the RT-PCR analysis to show the LRRFIP2 isoform mRNA levels, and the immunoblot analysis to show both endogenous and ectopically expressed LRRFIP2 isoform proteins using the flag-tagged antibody as well as the LRRFIP2 antibody (**Fig. D**). We have added these new results to our manuscript (Fig. 5e, and 6b).

Fig. D. Endogenous LRRFIP2 variant 2 and ectopic LRRFIP2 variants are detected using LRRFIP2 antibody. Immunoblot analysis of LRRFIP2 shows endogenous LRRFIP2 variant 2 and Flag-tagged variant 2 and 3.

6. In Fig 5, the authors identified CARM1 as a LRRFIP2 interacting partner and showed that V2 has higher affinity binding to CARM1. The authors then discussed how V3 expression can enhance CARM1 methylation activity on the SERPINE1 promoter. LRRFIP2 is located in cytoplasm, whereas CARM1 is in the nucleus. What is the functional connection between LRRFIP2 variants and CARM1?

Response: In previous studies, LRRFIP2 has been studied mainly in macrophages, in which LRRFIP2 was found to be localized in the cytoplasm where it assists the co-localization of NLRP2, ASC and F-actin (Burger D. et al. and Jin J. et al.). In response to Reviewer #1's question, we have examined the intracellular localizations of CARM1 and LRRFIP2 in gastric cancer cells. We found that both LRRFIP2 and CARM1 were localized in the nuclear fraction. CARM1 was only found in the nucleus, while LRRFIP2 was found in the cytoplasm and the nucleus (**Fig. E**). Thus, these results suggest that the intracellular localization of LRRFIP2 may be cell type- and context- dependent. We have added this new result to our manuscript (Supplementary Fig. S14).

Fig. E. CARM1 and LRRFIP2 variants are co-localized in the nucleus while LRRFIP2 variants are also detected in the cytoplasm. The fractionated cell lysates were

immunoblotted to show intracellular localization of CARM1 and LRRFIP2.

References

Burger, D., Fickentscher, C., de Moerloose, P. & Brandt, K.J. F-actin dampens NLRP3 inflammasome activity via Flightless-I and LRRFIP2. *Sci Rep* **6**, 29834 (2016).

Jin, J. et al. LRRFIP2 negatively regulates NLRP3 inflammasome activation in macrophages by promoting Flightless-I-mediated caspase-1 inhibition. *Nat Commun* **4**, 2075 (2013).

7. The authors stated that CARM1 directly regulates SERPINE1 transcription and showed that CARM1 binds the SERPINE1 promoter by ChIP. The authors are required to directly examine the transcription activity of CARM1 on the SERPINE1 promoter, for example, using promoter luciferase assays. It is necessary to experimentally demonstrate 1) CARM1 and ACTR directly regulate SERPINE1 transcription; 2) CARM1 and ACTR interaction is differentially affected by LRRFIP2 isoforms; and 3) CARM1 and ACTR are the direct downstream effectors of LRRFIP2 that regulate SERPINE1 transcription.

Response: As suggested by Reviewer#1, we have performed promoter luciferase assay following knockdown of CARM1 and ACTR to demonstrate that CARM1 and ACTR directly regulate *SERPINE1* transcription. Supporting previous results, knockdown of CARM1 and ACTR significantly reduced the promoter activity of *SERPINE1* (**Fig. F**), suggesting that the transcriptional regulation by CARM1 and ACTR is a direct event. These results have been added to Supplementary Fig. 16b and Fig. 6f.

Furthermore, we showed that exon 7 deletion decreased the interaction between ACTR and CARM1, whereas LRRFIP2 variant 3 overexpression increased the interaction between CARM1 and ACTR as shown in Figs 6a and 6b. Additionally, we demonstrated that overexpression of LRRFIP2 variant 2 decreased the interaction between CARM1 and ACTR as shown in Supplementary Fig. 19c (**Fig. G**). Together, these results support the notion that LRRFIP2 variants differentially regulate the interaction between CARM1 and ACTR, which further directly regulate *SERPINE1* transcription.

Fig. F. Knockdown of CARM1 or ACTR reduces the luciferase activity of the *SERPINE1* promoter. Control and exon 7 deleted MKN1 cells were transfected with *SERPINE1* promoter (-1500/+500) and then assayed for luciferase activity. The data represent the mean \pm SD of independent experiments. * $p < 0.05$; ** $p < 0.005$; *** $p < 0.0005$; n.s: not significant.

Fig. G. Overexpression of LRRFIP2 variant 2 decreased the interaction between CARM1 and ACTR. Immunoprecipitation analysis showing interaction between CARM1 and ACTR in the presence of LRRFIP2 variant 2.

Minor points:

1. Some of the invasion/migration assays and the western blots showed subtle differences. The authors should optimize experimental conditions before further conclusions can be

drawn.

- Fig 5i, Fig 6a, f: the differences are subtle

Response: We have repeated the experiments and observed more significant data (Fig. 5i became Fig. 5k in revised Fig).

- Fig 5d (CARM1 KD), Fig 6c (siGRIP1, siSRC1): the KD efficiency seems very poor, or the antibodies are not good

Response: We have repeated the experiments and substituted the results as shown in Fig. 5d.

- Fig 6b: the quality of the blots is not clear enough to tell whether any of the conclusions are reliable

Response for Fig. 6: CARM1 was originally identified as a binding partner of GRIP1/TIF2/Src-2/NCOA2, a member of the p160 family of steroid receptor coactivators, in a yeast two-hybrid screen, and other p160 family members (Src-1/NCOA1, ACTR/AIB1/SRC-3/NCOA3) were also shown to directly interact with CARM1 (Chen et al., 1999; Koh et al., 2001; Lee et al., 2005, Wsocka et al., 2006). Since the p160 coactivator family serves as a binding platform for CARM1, assisting its role as a coregulator of transcription, we investigated whether LRRFIP2 variant 2 suppresses the tumor promoting activity of CARM1 by abrogating the interaction between CARM1 and a member of the p160 family. Immunoprecipitation assay revealed that ACTR is the only p160 family member which strongly interacted with CARM1 in the gastric cancer cell lines used in this study. As shown in Fig. 6, we observed that exon 7 deletion in LRRFIP2 variant 3 led to reduced interaction between ACTR and CARM1 and knockdown of ACTR dramatically decreased the migration and invasion of MKN1 cells.

As suggested by Reviewer #1, we repeated immunoprecipitation assay. Our results show that endogenous expression levels of GRIP1 and SRC1 proteins are very low and that there is very weak or no interaction between GRIP1 or SRC1 and CARM1 in both MKN1 and MKN28 cell lines (**Fig. Ha** and **b**). However, only the ACTR results are included in Fig. 6 because it may confuse the readers. Also, the Western blots, which caused confusion, were replaced.

Fig. H. CARM1 shows very weak interaction with SRC1 in only MKN1 cells and no interaction with GRIP1 in MKN1 or MKN28 cells. Immunoprecipitation analysis showing interaction between CARM1 and GRIP1 or SRC1.

References

Chen, D. *et al.* Regulation of transcription by a protein methyltransferase. *Science* **284**, 2174-7 (1999).

Koh, S.S., Chen, D., Lee, Y.H. & Stallcup, M.R. Synergistic enhancement of nuclear receptor function by p160 coactivators and two coactivators with protein methyltransferase activities. *J Biol Chem* **276**, 1089-98 (2001).

Lee, D.Y., Teyssier, C., Strahl, B.D. & Stallcup, M.R. Role of protein methylation in regulation of transcription. *Endocr Rev* **26**, 147-70 (2005).

Wysocka, J., Allis, C.D. & Coonrod, S. Histone arginine methylation and its dynamic regulation. *Front Biosci* **11**, 344-55 (2006).

2. Several figures have p values and significance levels that are questionable, and a few figures have mistakes that needed proofreading:

- Fig 1d-e, Fig 2d-e, Fig 4b: the p value of the GO and KEGG pathways seem too big for analyzing a standard differentially expressed gene list.

Response: As Reviewer #1 pointed out, Fig. 1 e had an error regarding p-values; the x-axis of 1e was miswritten. The p-values of Supplementary Fig. 7a-b (previously Fig. 2d-e) and Fig. 4b are less than 0.05. The p-values for these figures were calculated using Enrichr web software.

- Fig 2m, Fig 5i: figures need proofreading

Response: We have corrected the figures (Fig. 2m and 5i are Fig. 2l and 5l in the revised Fig., respectively).

- Fig 2i-m, Fig 4e: the difference in TCGA subgroups is very subtle. The authors also did not explain why the sample numbers of all the box-and-whisker plots are different.

Response: We showed that the expression differences in TCGA subgroups are significant. (P-values are provided for each figure) The reason why the sample numbers are different is that we used TCGA for the gene expression while we used SpliceSeq for the isoform expression level. The SpliceSeq is a processed database originated from TCGA. For clarification, we changed the database name in the boxplots. We described this in Method Sections of the Text (p.20).

- Fig 7a-b: would suggest rerunning the samples

Response: We replaced the figure with more clear blots as suggested (Fig. 7a and 7b).

3. The authors have been using the exon 7 KO cell lines as a loss-of-function of LRRFIP2 V3. However, from Fig 3g, the KO is also a gain-of-function of V2. The authors should at least comment on the possibility that the exon 7 KO phenotypes could also be caused by the gain of V2 in addition to the loss of V3.

Response: We added a statement to the Result section that exon 7 deletion caused not only a loss-of-function of variant 3 but also a gain-of-function of variant 2 (p.9, line 233-236). Also, in order to clarify the effect of a gain-of-function of LRRFIP2 variant 2, apart from a loss-of-function of LRRFIP2 variant 3, we generated LRRFIP2 variant 2-overexpressing cell line using MKN1 cells. Interestingly, we could observe reduction of *SERPINE1* expression and H3R17 methylation when LRRFIP2 variant 2 was overexpressed in these cells (Fig. 1a and b). Also, binding affinity between CARM1 and ACTR was reduced following overexpression of LRRFIP2 variant 2 and reduction of migration and invasion ability was statistically significant as well (Fig. 1c and 1d). We have included these data in Supplementary Fig. 19.

Fig. 1. Overexpression of LRRFIP2 variant 2 in MKN1 cells reduces the metastatic potential of gastric cancer cells. a RT-PCR analysis showing mRNA level of *SERPINE1*. **b** Immunoblot analysis of Histone H3R17me2. **c** Immunoprecipitation assay showing the interaction between endogenous CARM1 and ACTR. **d** Transwell migration assay and Matrigel invasion assay of MKN28 cell lines upon overexpression of LRRFIP2 variant 2 (left) and bar graphs showing number of invaded and migrated cells (right), respectively, following staining with crystal violet.

4. “an exon 7-truncated form” should be stated as exon-7 skipped form

Response: We replaced “an exon -7-truncated form” with “an exon 7-skipped form” as suggested by Reviewer #1.

5. 1st paragraph 2nd line – “As in the previous study about ESRP1...”– cite the reference.

Response: We added the references in the second line of the first paragraph in the results section (p5, line 96-98).

6. Some of the Supplemental figures should be combined to pair with the main figures.

Response: We moved Fig.1j, Fig.3c to the main figures from supplementary data, and the sashimi plots were also added to Fig. 1h and Fig. 2f.

In conclusion, the authors showed strong evidence that the LRRFIP2 V3 promotes gastric cancer liver metastasis. Some of the data in this manuscript needs more refinement. Importantly, the authors are required to provide causal evidence to support the axis of ESRP1 – LRRFIP2 splicing – CARM1 methylation – SERPINE1 transcriptional activation – metastatic phenotypes.

Reviewer #2 (Remarks to the Author):

The manuscript by Lee et al entitled “ESRP1-regulated isoform switching of LRRFIP2 determines metastasis of gastric cancer” reported the identification and functional characterization of a novel ESRP1 RNA splicing target gene LRRFIP2 in gastric cancer. Mechanistically, the authors propose that variant 2 of LRRFIP2 competes with ACTR for CARM1 binding to suppress CARM1 activity to inhibit metastasis. The functional data of the manuscript is interesting with translational implications. However, the major weaknesses of the manuscript will have to do the mechanism of action and scientific rigor. The authors should consider the following comments to support their conclusions.

General comments

1. Mechanism of action: The notion for fig. 6 that LRRFIP2-v2 interacts with CARM1 to compete with ACTR is not convincing. The interaction between CARM1 and ACTR in fig. 6b was not decreased by overexpression of v2. And this notion cannot explain why overexpression of v3 increased the interaction between CARM1 and ACTR. The notion also cannot explain why overexpression of v2 did not affect invasion and migration while overexpression of v3 promoted invasion and migration. Instead, all the results in fig. 6 suggested that v3 promotes the interaction between CARM1 and ACTR and v2 variant has no effect on the interaction. This is consistent with fig. 6a that conversion of v3 to v2 by CRISPR KO of exon 7 decreased the expression of v3, and therefore resulted in a decreased interaction between CARM1 and ACTR. However, this still cannot explain Fig. 5b that v2 dominantly interacts with CARM1. Overall, the proposed mechanism of action cannot explain the phenotypes reported in this study.

In addition, perhaps more importantly, the authors need to explore how v3 promotes CARM1 activity (and possibly the interaction between CARM1 and ACTR) to mediate the metastasis-promoting effects of LRRFIP2 v3.

Response: We apologize for the confusion by not sufficiently explaining the roles of LRRFIP2 variant 2 and LRRFIP2 variant 3. To investigate whether LRRFIP2 variant 2 inhibits the binding of ACTR to CARM1 through direct binding to CARM1, we generated a cell line overexpressing LRRFIP2 variant 2 in MKN1 cells with high expression of LRRFIP2 variant 3. Interestingly, the binding of ACTR to CARM1 was markedly reduced by ectopic expression of LRRFIP2 variant 2 (**Fig. Aa**) (Supplementary Fig. 19a). In addition, overexpression of LRRFIP2 variant 2 in MKN1 cells downregulated the expression of SERPINE1, asymmetric dimethylation of histone H3R17, invasiveness and migratory potential (**Fig. Ab-d**) (Supplementary Fig. 19b-d).

In order to respond to Review #2's question, we also investigated how LRRFIP2 variant 3 enhances the metastasis-inducing activity of CARM1. We transiently transfected CARM1 with LRRFIP2 variants 2 or 3, and examined interaction between LRRFIP2 variant 2 and CARM1 in the presence or absence of LRRFIP2 variant 3. Surprisingly, LRRFIP2 variant 3 strongly interacted with LRRFIP2 variant 2, resulting in decreased interaction between

LRRFIP2 variant 2 and CARM1 (**Fig. Ae**). This may be how LRRFIP2 variant 3 enhances metastatic activity of CARM1. We have included these results in Supplementary Fig. 18 and 19.

Fig. A. Overexpression of LRRFIP2 variant 2 in MKN1 cells reduces the metastatic potential of gastric cancer cells. **a** Immunoprecipitation assay showing the interaction between endogenous CARM1 and ACTR. **b** RT-PCR analysis showing mRNA level of SERPINE1. **c** Immunoblot analysis of Histone H3R17me2. **d** Transwell migration assay and Matrigel invasion assay of MKN28 cell lines upon overexpression of LRRFIP2 variant 2 (left) and bar graphs showing number of invaded and migrated cells (right), respectively, following staining with crystal violet. **e** Immunoprecipitation assay showing the interaction between CARM1 and LRRFIP2 variant 2 in the presence of LRRFIP2 variant 3. The data represent the mean \pm SD of independent experiments. * $p < 0.05$; ** $p < 0.005$; *** $p < 0.0005$; n.s: not significant.

2. Along these lines, genome-wide unbiased approaches such as ChIP-seq and RNA-seq could be helpful in addressing the missing mechanistic links among CARM1, LRRFIP2 v2 vs. v3, and ACTR.

Response: As Reviewer#2 has commented, we tried to generate CARM1 KO and ACTR KO using CRISPR-Cas9 system in MKN1 cells. We successfully transfected Lenti-CRISPR v2 vector targeting CARM1 and ACTR in MKN1 cells (**Fig. Ba** and **b**). However, when single-

cell clones were grown in 96 well plates, it showed morphological features of cell death and most cells that were further cultured in 24 and 12 well plates underwent apoptosis (**Fig. Bc** and **d**). Thus, we instead selected the genes that were significantly regulated in both of the knockout clones and known to function in cancer progression and metastasis from Fig. 4a, and examined their expression following knockdown of CARM1 and ACTR using siRNAs (**Fig. C**). Expression of several genes such as *CERK*, *PKP1* and *COL5A2* was reduced by knockdown of CARM1 and ACTR, suggesting the possibility that these genes could also be transcriptionally regulated by CARM1 and ACTR along with LRRFIP2 variant switch, like *SERPINE1*. Therefore, we intend to deepen our investigation by examining the transcriptional regulation of the genes for further study.

Fig. B. CARM1 and ACTR knockout cell lines failed to grow after single cell selection. a,b T7E1 assay was conducted following Lenti-CRISPR v2 vectors targeting CARM1 and ACTR. **c, d** Microscopic image shows CARM1 and ACTR KO cells after single cell selection.

Fig. C. Knockdown of CARM1 or ACTR mediates transcriptional regulation of the target genes validated by RNA-seq. qRT-PCR analysis of genes that were downregulated by exon 7 KO following knockdown of CARM1 and ACTR. All P values were calculated by unpaired two-tailed Student's t tests. These data represent the mean \pm S.D. *p<0.05; **p<0.005; ***p<0.0005; n.s.: not significant.

3. The ChIP analysis (should use qPCR analysis) and RT-qPCR analysis should be quantified with at least three biological repeats (agarose gel imaging is not sufficient in this day and age).

Response: As suggested by Reviewer #2, we replaced the agarose gel imaging with qPCR analysis for Fig. 5h, 5i, 6d, 6e, Fig.7c, 7d, 7e and Supplementary Fig. 17a. We could not change other RT-PCR results such as Fig. 1k, Supplementary Fig.6, Fig.2g, Fig. 3b, c, h, Fig.5g, Fig.7f since LRRFIP2 variants are indistinguishable by RT-qPCR; the difference between the two variants is only the presence or absence of exon 7.

4. For functional studies (at least for in vitro), please include more than one cell line to limit potential cell line specific effects.

Response: We established exon 7 knockout cell lines using SNU484 and LRRFIP2 variants 2

and 3-overexpressing cell lines using MKN74 to conduct functional studies. Consistent with MKN28 cells, overexpression of LRRFIP2 variant 3 induced expression of *SERPINE1* and methylation of H3R17, which contributed to increased invasion and migration of MKN74 cells (**Fig. D**). Also, exon 7-knockout cell lines were made with SNU484 cells, which showed consistent results as MKN1 cells. Again, knockout of exon 7 led to reduction of *SERPINE1* expression, histone H3R17 methylation, and invasiveness and migratory potential (**Fig. E**). These results were added to Supplementary Fig. 10 and 11.

Fig. D. Overexpression of LRRFIP2 variant 3 increases invasiveness and migration of MKN74 cells. **a** RT-PCR analysis of *LRRFIP2* and *SERPINE1* in MKN74 cells overexpressing variants 2 and 3. **b** Immunoblot analysis of Flag-tagged LRRFIP2 and histone H3R17 methylation. **c** Transwell migration assay and Matrigel invasion assay of LRRFIP2 variants 2 and 3-overexpressing MKN74 cells (left) and bar graphs showing number of invaded and migrated cells (right), respectively, following staining with crystal violet. All P values were calculated by unpaired two-tailed Student's t tests. These data represent the mean \pm S.D. * $p < 0.05$; ** $p < 0.005$; *** $p < 0.0005$; n.s: not significant.

Fig. E. Knockout of LRRFIP2 exon 7 reduces metastatic potential of SNU484 cells. **a** RT-PCR analysis of *LRRFIP2* and *SERPINE1* in exon 7-deleted SNU484 cells. **b** Immunoblot analysis of LRRFIP2 and histone H3R17 methylation in exon 7-deleted SNU484 cells. **c** Transwell migration assay and Matrigel invasion assay of exon 7-deleted cell lines (left) and bar graphs showing number of invaded and migrated cells (right), respectively, following staining with crystal violet. All P values were calculated by unpaired two-tailed Student's t tests. These data represent the mean \pm S.D. * $p < 0.05$; ** $p < 0.005$; *** $p < 0.0005$; n.s: not significant.

Specific comments:

1. Fig. 2a, please specify how the cut-off was determined for Fig. 2b and 2c.

Response: We subtracted Pearson's correlation coefficient of the expression of *ESRP1* and the 2nd set of isoforms from Pearson's correlation coefficient of the expression of *ESRP1* and the 1st set of isoforms. Then we neglected the genes whose sum of the average of the relative expression of isoform 1 and isoform 2 is too small (less than 0.5). The top 20 genes in the resulting list were used to generate the heatmap in Fig. 2 and c. We added this description in the manuscript.

2. Fig. 2i-l. The correlation analysis between *ESRP1* expression and *LRRFIP2v3* in TCGA dataset is not proper. The numbers of TCGA subgroups are different between *ESRP1* expression and *LRRFIP2v3* PSI groups. Since they are in the same TCGA cancer type, the

numbers of each subgroup should be the same. Correlation should be done in the same sample pool.

Response: The reason why the sample numbers are different is that we used TCGA for the gene expression while we used SpliceSeq for the isoform expression level (Ryan et al., 2012). The SpliceSeq is a processed database originated from TCGA. For clarification, we changed the database name in the boxplots.

Reference

Ryan, M.C., Cleland, J., Kim, R., Wong, W.C. & Weinstein, J.N. SpliceSeq: a resource for analysis and visualization of RNA-Seq data on alternative splicing and its functional impacts. *Bioinformatics* **28**, 2385-7 (2012).

3. In fig. 2m, is ESRP1 expression positively correlated with survival in this dataset?

Response: We couldn't detect a statistically significant correlation between *ESRP1* expression and patient survival in the dataset we used in Fig. 2l (previously Fig. 2m), possibly due to heterogenic characteristic of gastric cancer. It is widely appreciated that gastric cancer cells are characterized with extensive intertumoral and intratumoral heterogeneity (Cancer Genome Atlas Research, 2014; Wang et al., 2021; Wang et al., 2020). Even our tissue data in Fig. 2b and c shows more diversity compared to the cell line data regarding correlation between the expression of splicing variants and *ESRP1*. Moreover, patients whose data were analyzed in Fig. 2l (previously Fig. 2m) had measurable and histologically or cytologically confirmed metastatic and/or recurrent gastric adenocarcinoma (Kim et al., 2021). Thus, it may not display a significant correlation between *ESRP1* and patient survival. Further investigations such as single cell transcriptome profiling in gastric cancer tissue would be needed to better understand the correlation between expression of *ESRP1* and its target splicing variants and patient survival.

1

References

Cancer Genome Atlas Research, N. Comprehensive molecular characterization of gastric adenocarcinoma. *Nature* **513**, 202-9 (2014).

Kim, S.T. et al. Comprehensive molecular characterization of gastric cancer patients from phase II second-line ramucirumab plus paclitaxel therapy trial. *Genome Med* **13**, 11 (2021).

Wang, R. et al. Single-cell dissection of intratumoral heterogeneity and lineage diversity in metastatic gastric adenocarcinoma. *Nat Med* **27**, 141-151 (2021).

Wang, R. et al. Multiplex profiling of peritoneal metastases from gastric adenocarcinoma identified novel targets and molecular subtypes that predict treatment response. *Gut* **69**, 18-31 (2020).

4. Fig. 4f and Supplementary Fig. 12b used auto select best cutoff function in Kaplan-Meier plotter for the survival plot, which is not proper. Median or quartile split of patients is more objective.

Response: As reviewer#2 has suggested, we replaced the survival plot with a quartile split of patients (Fig. F). This has been added to Fig. 4f and Supplementary Fig 13.

Fig. F. High expression of *SERPINE1* and *CARM1* is associated with poor overall survival of gastric cancer patients. Kaplan-Meier analysis showing relapse-free survival depending on **a** *SERPINE1* and **b** *CARM1* expression levels from public meta-analysis data (N=875). The patients were split by lower quartile. P values were calculated from log-rank test.

5. Fig. 5, please provide evidence of interaction between *CARM1* and *LRRFIP2* v2 and v3 at the endogenous level.

Response: We conducted an immunoprecipitation assay to examine the interaction between *CARM1* and *LRRFIP2* variant 2 and 3 at the endogenous level. However, bands from IgG heavy chain interfere with those of endogenous *LRRFIP2* variants (Fig. G). Thus, we performed an immunoprecipitation assay with Flag-tagged *LRRFIP2* variants and endogenous *CARM1* instead, to further support the data of the interaction between ectopically expressed *LRRFIP2* variants and *CARM1* (Fig. 5c).

Fig. G. IgG heavy chain completely covers the endogenous LRRFIP2 variants. Immunoprecipitation assay conducted to show endogenous interaction between CARM1 and LRRFIP2.

Please quantify Fig. 5k by using H score. In addition, IHC against different targets such as H3R17me2a and SERPINE1 should be performed in serial sections and pictures should be taken in the roughly same regions for comparison. The pictures seem to be taken in random regions.

Response: We have replaced the IHC pictures taken in the same region and quantified the expression of SERPINE1 and H3R17me2 in Fig. 5j (previously Fig.5k) and added to the figure as suggested (**Fig. H**).

Fig. H. Expression of SERPINE1 and the dimethylation of H3R17 were significantly decreased in the liver tissues of the mice injected with exon 7-deleted cells.

Representative IHC images showing SERPINE1 and H3R17me2a expression in metastasized liver tissues from **Fig.3j** and **k**. Original magnification, $\times 200$. Scale bar, 50 μ m.

6. Fig. 5g, Overexpression of V3-LRRFIP2 should not affect the endogenous level of V2-LRRFIP2. Thus, a lower band of V2-LRRFIP2 should be observed.

Response: As the reviewer mentioned, it is definitely questionable how the lower band representing endogenous LRRFIP2 variant 2 disappears when variant 3 is overexpressed. Since we could not detect the endogenous LRRFIP2 with the antibody previously, we have ordered a new LRRFIP2 antibody (abcam, ab172367) and successfully observed the protein expression (**Fig. I**). Immunoblot analysis shows that endogenous LRRFIP2 is intact even

when ectopic variants are expressed. Thus, the seemingly reduced expression of endogenous *LRRFIP2* variant 2 may be due to limitation of RT-PCR analysis, which preferentially amplifies the cDNA with greater concentration. Protein expression of endogenous *LRRFIP2* variants have been added to Fig. 5e, f, 6a and b.

Fig. I. The expression of endogenous *LRRFIP2* variant 2 remains unchanged when ectopic *LRRFIP2* variants are transfected. Immunoblot and RT-PCR analysis of *LRRFIP2* showing endogenous *LRRFIP2* variant 2 and Flag-tagged variant 2 and 3.

7. *The conclusion for Fig. 5 is not proper. There is no evidence suggest that LRRFIP2-v3 is critical for CARM1 enzymatic activity. Fig. 5 suggests that LRRFIP2-v3 affect the recruitment of CARM1 to target genes.*

Response: We made the conclusion that “*LRRFIP2* variant 3 is critical for *CARM1* enzymatic activity” since knockout of exon7 reduces histone H3R17 methylation while overexpression of *LRRFIP2* variant 3 induces it. However, we agree that further experiments are needed to support that notion. Thus, we replaced the conclusion with “*LRRFIP2* variant 3 may assist the recruitment of *CARM1* to target genes, regulating the transcriptional activation of metastasis-promoting genes such as *SERPINE1*” (p13, line 351-353).

8. *The data as presented in Fig. 6a do not appear to support the conclusion that exon 7 knockout only affects CARM1’s interaction with ACTR (as both GRIP1 and SRC1 appear to show a similar decrease in interaction). Likewise, in Fig. 6c, GRIP1 and SRC1 knockdown also reduced expression of SERPINE1. In addition, 6a anti-GRIP1 and anti-SRC1 IP blots look similar. Anti-GRIP1 IP immunoblot in 6b was not successful. In 6a, SRC1 and GRIP1 were detected as a single band in WCL but two bands in IP blots and 6b immunoblots. Immunoblots for CARM1 are also not consistent with a single band in 6a and 6b IP while two bands in 6b WCL. Please clarify.*

Response: CARM1 was originally identified as a binding partner of GRIP1/TIF2/Src-2/NCOA2, a member of the p160 family of steroid receptor coactivators, in a yeast two-hybrid screen, and other p160 family members (Src-1/NCOA1, ACTR/AIB1/SRC-3/NCOA3) were also shown to directly interact with CARM1 (Chen et al., 1999; Koh et al., 2001; Lee et al., 2005, Wysocka et al., 2006). Since the p160 coactivator family serves as a binding platform for CARM1, assisting its role as a coregulator of transcription, we investigated whether LRRFIP2 variant 2 diminishes the activity of CARM1 by abrogating the interaction between CARM1 and a member of the p160 family. Immunoprecipitation assay revealed that ACTR is the only p160 family member which strongly interacted with CARM1 in the gastric cancer cell lines used in this study.

The previous Fig. 6a caused confusion because we reblotted the immunoprecipitation bands with GRIP1 and SRC1 after blotting with ACTR. The repeated immunoprecipitation assay shows very weak or no interaction between GRIP1 or SRC1 and CARM1 in both MKN1 and MKN28 cell lines (**Fig. Ja** and **b**). Also, the expression of the two proteins is much weaker compared to that of ACTR, suggesting that among the p160 family members ACTR might mainly function to assist CARM1 regulating transcription in these cell lines. We tried to support the notion that only ACTR is important regarding interaction with CARM1 which is affected by variant switch of LRRFIP2 in this context. To avoid confusion, we included only the results of ACTR in Fig. 6 and the Western blots that caused confusion were replaced.

Fig. J. CARM1 shows very weak interaction with SRC1 in only MKN1 cells and no interaction with GRIP1 in MKN1 or MKN28 cells. Immunoprecipitation analysis shows interaction between CARM1 and GRIP1 or SRC1.

References

Chen, D. *et al.* Regulation of transcription by a protein methyltransferase. *Science* **284**, 2174-7 (1999).

Koh, S.S., Chen, D., Lee, Y.H. & Stallcup, M.R. Synergistic enhancement of nuclear receptor function by p160 coactivators and two coactivators with protein methyltransferase activities. *J Biol Chem* **276**, 1089-98 (2001).

Lee, D.Y., Teyssier, C., Strahl, B.D. & Stallcup, M.R. Role of protein methylation in regulation of transcription. *Endocr Rev* **26**, 147-70 (2005).

Wysocka, J., Allis, C.D. & Coonrod, S. Histone arginine methylation and its dynamic regulation. *Front Biosci* **11**, 344-55 (2006).

9. The description for Fig. 7c-d is not proper. CARM1 inhibitor EZM2302 did not affect the recruitment of CARM1 to SERPINE1 promoter.

Response: We agree that the description for Fig. 7c and d was not correct, as Reviewer #2 has pointed out. The effect of EZM2302 on the CHIP assay was observed only when LRRFIP2 variant 3 was highly expressed. Thus, “Notably, inhibition of the enzymatic activity of CARM1 reduced both the recruitment of CARM1 and the enrichment of the asymmetric methylation of histone H3R17 on the *SERPINE1* promoter, which was enhanced by overexpression of LRRFIP2 variant 3 (Fig. 7c and d)” was replaced with “Notably, inhibition of the enzymatic activity of CARM1 reduced both the recruitment of CARM1 and the enrichment of the asymmetric methylation of histone H3R17 on the *SERPINE1* promoter only when LRRFIP2 variant 3 was expressed (Fig. 7c and d).”

10. What is the expression/amplification status of ESRP1 in gastric cancer in TCGA?

Response: In our study, since the focus is on the correlation between the expression levels of LRRFIP2 variants and ESRP1 and how they functionally regulate the metastatic potential of gastric cancer cells, the expression status of ESRP1 was compared between stages of cancer and metastatic/non-metastatic cancer types as in Fig. 2h-k. We observed a significantly lower expression level of *ESRP1* and a higher expression level of *LRRFIP2* variant 3 in more advanced gastric cancer stages (Fig. 2h and i), and significantly lower mRNA expression of *ESRP1* and higher mRNA expression of *LRRFIP2* variant 3 in more metastatic gastric cancer (Fig. 2j and k).

We also analyzed the amplification status of ESRP1 in TCGA, as suggested (**Fig. K**). Although the amplification frequency in gastric cancer is not relatively very low compared to

other types of cancers, the amplification frequency of ESRP1 in gastric cancer is not necessarily required to support our theory in this study. Instead, these data would improve our understanding of the role of ESRP1 in cancer metastasis if we could analyze the alteration frequency of ESRP1 expression in metastatic and non-metastatic samples,

Fig. K. ESRP1 has varying alteration frequency in different types of tumor in TCGA. A bar graph shows frequency of genomic alterations (mutation, fusion, amplification, deep deletion, multiple alterations) of ESRP1 across different tumor types.

11. Scale bars and molecular weight markers are missing for all the figures.

: We have added the scale bars and molecular weight markers as Reviewer #2 suggested.

Reviewer #3 (Remarks to the Author):

The study investigates the effect of different variants of LRRFIP2 in gastric cancer metastasis. The authors argue that LRRFIP2 variant 3 but not variant 2 induces metastasis via coactivating transcription of SERPINE1 with CARM1. The authors also argue that LRRFIP2 variant 3 is present mostly in tumors with low expression of ESRP1, as ESRP1 is responsible for processing LRRFIP2 variant 3 into variant 2, and low ESRP1 expression or high LRRFIP2 variant 3 correlates with poor patient survival. The study intriguingly argue that different variants might function differently, which has been often overlooked. However, a few major concerns remain for this investigation.

1. There is no analysis on the protein levels of the two LRRFIP2 variants. Only RT-PCR assessing the mRNA levels was shown. Since one exon is deleted in the variant 2, there should be a change of molecular weight which should be detectable in western blot. Also, for the RT-PCR measure mRNA levels of these variants, the authors should provide a schematic showing how the primers flank the regions of different exons; and the primer sequences should be provided in the Methods.

Response: As suggested by Reviewer#3, we performed Western blot analysis to investigate the protein levels of the two LRRFIP2 variants. As shown in **Fig. A** below, we observed the protein expression of the two variants. We have added these new results to our manuscript (Figure 5e, 5f, 6a, and 6b).

In addition, a schematic diagram showing how the primers flank the exons of the two variants to produce PCR products of two different sizes is added to Fig. 1J. Primer sequences are described in the Materials and Methods (p.20) (**Fig. B**).

Fig. A. Exon 7 knockout of LRRFIP2 is detected in protein and mRNA. Immunoblot and RT-PCR analysis showing LRRFIP2 exon 7 knockout.

Fig. B. The Same primer set detects LRRFIP2 variant 2 and 3, producing PCR products with two different sizes. A schematic diagram shows how mRNAs of LRRFIP2 variant 3 and 2 are detected by PCR.

2. The CRISPR approach deleting exon 7 of LRRFIP2 is doubtful. Again, a western blot analysis is critical to make the judge whether only LRRFIP2 v3 is knockout leaving v2 intact. Assessing mRNA with RT-PCR is not a way to measure CRISPR knockout. The authors could also sequence the knockout clones and show the mutations generated by the knockout. CRISPR knockouts a gene by generating premature stop codon, so probably both v2 and v3 have been knockout. To target v3 only, the authors can consider using shRNAs specifically targeting exon.

Response: We used two target sites to specifically knockout exon 7 (gRNA1: 5'-CCTCCATATATAGCCC TGTCCCC-3'; gRNA2: 5'-CCGTGGTGTCTTAGCCATACAAA-3'). (**Fig. Ba**). Oligonucleotides were synthesized and ligated into pSpCas9(BB)-2A-Puro (PX459) as previously reported (Ran et al., 2013). The cells were transfected with the vectors containing gRNA sequence using the Neon Transfection System (Thermo Fisher Scientific). After clonal selection, deletion of exon 7 was determined by sequencing and RT-PCR, and further confirmed by RNA-sequencing in Fig. 4a(**Fig. Bb and Bc**). Clone 1-1 and 1-4 were used as KO#1 and KO#2 for further experiments.

b Sequencing result after clonal selection

Control	GCTTTTTCTCCATATATAGCCTGTCCC...exon7...TGGTGTCTTAGCCATACAAAATTAAA
clone1-1	GCTTTTTCTCCA-----TGTCTTAGCCATACAAAATTAAA
clone1-4	GCTTTTTCTCCA-----TGTCTTAGCCATACAAAATTAAA
clone2-1	GCTCCTTCATAGA-----TGTCTTAGCCATACAAAATTAAA
clone2-2	GCTTTTTCTCCA-----TGTCTTAGCCATACAAAATTAAA
clone2-4	GCTTTTTCTCCA-----TGTCTTAGCCATACAAAATTAAA
clone2-5	(*clone 2-5 was very noisy)

Fig. B. Knockout cell lines were generated using CRISPR/Cas9 system. a Two target sites of CRISPR/Cas9 designed to excise exon7. **b** Sequencing result of the clones after clonal selection. **c** RT-PCR analysis of LRRFIP2 in the knockout clones. 2% agarose gel was used to separate the bands of variants (294bp and 222bp). Red box indicates the clones used in the experiments.

Reference

Ran, F.A. et al. Genome engineering using the CRISPR-Cas9 system. *Nat Protoc* **8**, 2281-2308 (2013).

3. The authors should provide in vivo data for the drug targeting CARM1.

Response: As suggested by Reviewer#3, we performed in vivo liver metastasis experiment using CARM1 inhibitor, EZM2302. From seven days after intrasplenic injection, EZM2302 or vehicle (0.5% methylcellulose in distilled water) was administered orally BID at a dose of 100 mg/kg for 21 days according to references (Drew et al., 2017; Greenblatt et al., 2018). Five weeks after the intrasplenic injection, the mice were sacrificed and the livers were removed and prepared for histological examination (hematoxylin and eosin (H&E) and IHC (Immunohistochemistry) staining). Liver nodules were dramatically reduced when EZM2302 was treated especially when LRRFIP2 variant 3-overexpressing cells were injected (**Fig. Ca**). In addition, a decrease in SERPINE1 expression and H3R17 methylation was observed by EZM2302 treatment. These results were added to Fig. 7i and j.

whole liver image showing metastatic nodules (left) and scatter plot showing the number of liver metastatic nodules (right). **b** Representative IHC images showing SERPINE1 and H3R17me2a expression in metastasized liver tissues from **a**. Original magnification, $\times 200$. Scale bar, 50 μm .

Reference

Drew, A.E. et al. Identification of a CARM1 Inhibitor with Potent In Vitro and In Vivo Activity in Preclinical Models of Multiple Myeloma. *Sci Rep* 7, 17993 (2017).

Greenblatt, S.M. et al. CARM1 Is Essential for Myeloid Leukemogenesis but Dispensable for Normal Hematopoiesis. *Cancer Cell* 34, 868 (2018).

4. Cell line models are scarce for the functional part in this study, with just one for overexpression and one for KO. The authors should try to perform knockout or knockdown in two independent cell lines.

Response: As suggested by Reviewer #3, we further established an overexpressing cell line using MKN74 and an exon 7 knockout cell line using SNU484 to perform functional studies. Consistent with MKN28 cells, overexpression of LRRFIP2 variant 3 induced expression of *SERPINE1* and methylation of H3R17, which seem to increase invasion and migration of MKN74 cells (**Fig. D**). Also, exon 7-knockout cell lines were made in SNU484 cells, which showed similar results as MKN1 cells. Again, knockout of exon 7 led to reduction of *SERPINE1* expression, histone H3R17 methylation, and invasiveness and migratory potential (**Fig. E**). These results were added to Supplementary Fig. 10 and 11.

Fig. D. Overexpression of LRRFIP2 variant 3 increases invasiveness and migration of MKN74 cells. **a** RT-PCR analysis of *LRRFIP2* and *SERPINE1* in MKN74 cells overexpressing variants 2 and 3. **b** Immunoblot analysis of Flag-tagged LRRFIP2 and histone H3R17 methylation. **c** Transwell migration assay and Matrigel invasion assay of LRRFIP2 variants 2 and 3-overexpressing MKN74 cells (left) and bar graphs showing number of invaded and migrated cells (right), respectively, following staining with crystal violet.

Fig. E. Knockout of LRRFIP2 exon 7 reduces metastatic potential of SNU484 cells. a RT-PCR analysis of *LRRFIP2* and *SERPINE1* in exon 7-deleted SNU484 cells. **b** Immunoblot analysis of LRRFIP2 and histone H3R17 methylation in exon 7-deleted SNU484 cells. **c** Transwell migration assay and Matrigel invasion assay of exon 7-deleted cell lines (left) and bar graphs showing number of invaded and migrated cells (right), respectively, following staining with crystal violet.

Apart from these major concerns, there are also a couple of minor ones.

1. The authors could consider providing metastasis free survival apart from overall survival as the investigation is mostly for metastasis.

Response: The patients whose data were analyzed in Fig. 21 (previously Fig. 2m) had measurable and histologically or cytologically confirmed metastatic and/or recurrent gastric adenocarcinoma (Kim et al., 2021). Therefore, the survival curves reflect metastases data and also include important status data that increase the clinical significance of the ESRP1 expression. The detailed description of the analysis can be found in the Materials and Methods section.

For other overall survival graphs using public data, metastasis-free survival of gastric cancer patients could not be obtained due to the limitations of public data of gastric cancer compared to other cancers. However, previous studies have shown that 30~60% of gastric cancer patients relapse with metastases rather than local recurrence and are often associated with death. Therefore, overall survival graphs reflect metastasis-free survival (Liu et al., 2016;

Mokadem et al., 2019; Spolverato et al., 2014).

References

Kim, S.T. et al. Comprehensive molecular characterization of gastric cancer patients from phase II second-line ramucirumab plus paclitaxel therapy trial. *Genome Med* **13**, 11 (2021).

Liu, D. et al. The patterns and timing of recurrence after curative resection for gastric cancer in China. *World J Surg Oncol* **14**, 305 (2016).

Mokadem, I. et al. Recurrence after preoperative chemotherapy and surgery for gastric adenocarcinoma: a multicenter study. *Gastric Cancer* **22**, 1263-1273 (2019).

Spolverato, G. et al. Rates and patterns of recurrence after curative intent resection for gastric cancer: a United States multi-institutional analysis. *J Am Coll Surg* **219**, 664-75 (2014).

2. The authors could start with tumor tissue data (fig 2) instead of cell line data (fig 1) showing LRRFIP2 and ESRP1 as cell line data becomes supportive when tumor tissue data is available.

Response: We appreciate your suggestion to start with Fig.2 instead of Fig.1. We also considered starting with tumor tissue data, as the correlation between the splicing variants of LRRFIP2 and ESRP1 seemed to be relatively more significant in the tumor tissues. However, a limitation was that we could not precisely divide tumor tissues into ESRP1-low and ESRP1-high groups as in cell lines. ESRP1 was present in all tumor tissue samples and its expression level gradually increases when sorted. Therefore, we concluded that presenting a heatmap of cell line data would be visually more robust than a heatmap of tumor tissue data.

REVIEWER COMMENTS

Reviewer #1 (Remarks to the Author):

The reviewer appreciates that the authors provided additional data to support their work. However, most of the comments from this reviewer were not addressed appropriately. It remains unclear whether ESRP1 directly binds LRRFIP2 and promotes exon 7 skipping. This is an important question as indicated by the title "ESRP1-regulated...". While the authors showed additional data that ESRP1 overexpression promotes v2 production, how ESRP1 affects LRRFIP2 splicing still remains unclear. Similarly, although the authors provided large amounts of data on experimentally manipulating the levels of LRRFIP2, CARM1, SERPINE1, the mechanistic studies on how LRRFIP2 isoforms affect CARM1 and how CARM1 acts on SERPINE1 still remain underexplored. Most of the work are correlative and phenotypic observations. Moreover, the figures are not well consolidated. As an example pointed out by this reviewer previously, Fig. 1 and Fig. 2 described very similar conclusions on ESRP1 and LRRFIP2's relationship, while Fig. 1 was based on cell line data and Fig. 2 was based on patient data. Additionally, the authors avoided using PSI (percent spliced in) to describe LRRFIP2, making it difficult to understand splice isoform switching. For instance, it is confusing to plot both v2 and v3 from the same sample in relationship with ESRP1 expression (Fig. 1i, 2f). At this stage, the reviewer finds that the findings are interesting, yet there are technical and conceptual issues. The conclusions and depth of study require extensive improvement.

Reviewer #2 (Remarks to the Author):

This reviewer thanks the authors for the tremendous efforts in addressing the concerns raised in the initial review. Those concerns have now been satisfactorily addressed either experimentally or by reasonable explanation. As such, the manuscript is now acceptable for publication. Rugang Zhang

Reviewer #3 (Remarks to the Author):

The revised version is much improved compared to the original. The authors have addressed the concerns from the reviews well. The only minor concern that remains to me is that the LRRFIP2 knockout, KO #1 and #2 in the figures are independent clones rather than clones from independent sequences, as clarifying in the rebuttal letter. This information should be stated clearly in the manuscript not just in the rebuttal. Since no independent sequence of gRNA has been used, to rule out off target effects from the gRNAs it shall be suggested to perform overexpression of LRRFIP2 v3 in at least one of the KO clones to rescue the effect from the KO.

ESRP1-regulated isoform switching of LRRFIP2 determines metastasis of gastric cancer

#NCOMMS-21-22414B

Response to Reviewers

We greatly appreciate the constructive comments from the reviewers and the invitation from the editor to submit a revised version. We have thoroughly revised the manuscript following the reviewers' suggestions. Please see below our point-to-point responses in non-italic text following reviewer comments in italic text.

Reviewer #1 (Remarks to the Author):

It remains unclear whether ESRP1 directly binds LRRFIP2 and promotes exon 7 skipping. This is an important question as indicated by the title "ESRP1-regulated...". While the authors showed additional data that ESRP1 overexpression promotes v2 production, how ESRP1 affects LRRFIP2 splicing still remains unclear.

Response:

To demonstrate that ESRP1 directly promotes exon 7 skipping of LRRFIP2, we performed RNA-immunoprecipitation assay (RIP) with anti-ESRP1 antibody in MKN28 cell line with high level of ESRP1 (Manco et al., 2021).

As recent studies demonstrated that ESRP binding motifs are often observed upstream of ESRP-silenced exons, we screened several regions upstream and downstream of exon 7 of LRRFIP2 variant 3 (Warzecha et al., 2010). Consistent with previously published data, RT-PCR analysis of the ESRP1 binding transcript revealed binding of ESRP1 to the upstream of the skipped exon in the presence of ESRP1 (**Fig. A**). In addition, after sequencing the purified PCR bands, we obtained the correct target sequence indicated by the grey box in **Figure Aa**. Together with **Figure 2b**, our results suggest that the interaction between ESRP1 and *LRRFIP2* transcript is direct and that the interaction induced by overexpressing ESRP1 in ESRP1-low cells silences exon 7 of LRRFIP2 variant 3. These results are shown in **Fig. 2c** and **Fig. S9** of the Supplementary Manuscript. Moreover, we obtained the exact target sequence marked in grey box in **Fig. Aa** following sequencing of the purified PCR band in **Fig. Ac**. Along with **Fig. 2b**, our result suggests that the interaction between ESRP1 and LRRFIP2 transcript is direct, and that ESRP1 overexpression in ESRP1-low cells silences exon 7 of LRRFIP2 variant 3. This result is added to **Fig. 2c** and **Supplementary Fig. S9** in the manuscript.

Fig. A. ESRP1 directly binds to the LRRFIP2 transcript in MKN28 cells. **a** Schematic representation and nucleotide sequence of the ESRP1 binding site in *LRRFIP2* gene. The red box represents exon 7 of LRRFIP2 variant 3 and the grey box represents the target sequence in **c**. **b** Western blot analysis of ESRP1 immunoprecipitation. **c** RT-PCR analysis of ESRP1- and IgG-bound transcripts following RNA-immunoprecipitation (RNA-IP).

References

- Manco, M. et al. The RNA-Binding Protein ESRP1 Modulates the Expression of RAC1b in Colorectal Cancer Cells. *Cancers (Basel)* 13(2021).
- Warzecha, C.C. et al. An ESRP-regulated splicing programme is abrogated during the epithelial-mesenchymal transition. *EMBO J* 29, 3286-300 (2010).

Similarly, although the authors provided large amounts of data on experimentally manipulating the levels of LRRFIP2, CARM1, SERPINE1, the mechanistic studies on how LRRFIP2 isoforms affect CARM1 and how CARM1 acts on SERPINE1 still remain underexplored. Most of the work are correlative and phenotypic observations.

Response:

We found CARM1 as a novel binding partner of LRRFIP2 variant 2, whose methylation activity was downregulated when exon 7 of variant 3 was deleted (**Fig. 4a-f**). In order to examine the mechanism of transcriptional regulation of *SERPINE1* by CARM1, along with the LRRFIP2 variants, we identified ACTR as a transcriptional coactivator which interacts with CARM1 and observed that expression of variant 2 reduces interaction between CARM1 and ACTR by endogenous immunoprecipitation assay (**Fig. 5a-b**), suggesting that direct binding of variant 2 to CARM1 prevents ACTR from binding to CARM1. We described this observation on **page 13, line# 309-310**.

As Reviewer #1 requested further explanation about how *SERPINE1* transcription is regulated, we conducted qRT-PCR assay in control MKN1 cells and exon 7-deleted MKN1 cells following overexpression of CARM1 and ACTR. As shown in **Fig. B**, *SERPINE1* expression was significantly lower in exon7-deleted MKN1 cells compared to MKN1 cells. In control cells, overexpression of CARM1 alone did not significantly increase *SERPINE1* expression, whereas simultaneous overexpression of CARM1 and ACTR increased transcription of *SERPINE1*. However, this increase was not significant in exon7-deleted MKN1 cells. On the other hand, CARM1 overexpression induced *SERPINE1* expression in exon 7-deleted MKN1 cells. Our data suggests that LRRFIP2-variant 2 generated by deletion of variant 3 inhibits recruitment of CARM1 in combination with the p160 family of coactivators onto the *SERPINE1* promoter, leading to decreased expression of *SERPINE1*. This result was added to **Fig. 5h**.

Fig. B. Transcription of *SERPINE1* is regulated by LRRFIP2 variants, CARM1 and ACTR. qRT-PCR result shows the relative mRNA expression of **a** *SERPINE1*, **b** *CARM1*, and **c** *ACTR*.

Moreover, the figures are not well consolidated. As an example pointed out by this reviewer previously, Fig. 1 and Fig. 2 described very similar conclusions on *ESRP1* and *LRRFIP2*'s relationship, while Fig. 1 was based on cell line data and Fig. 2 was based on patient data.

Response:

Fig.1 and **Fig.2** were rearranged according to Reviewer#1's suggestion. We placed all duplicate data between **Fig. 1** and **Fig. 2** in the **Supplementary Figures**.

Additionally, the authors avoided using PSI (percent spliced in) to describe *LRRFIP2*, making it difficult to understand splice isoform switching. For instance, it is confusing to plot both v2 and v3 from the same sample in relationship with *ESRP1* expression (Fig. 1i, 2f).

Response:

We agree that the rationale behind using the relative abundance (or frequency) of the TPM has not been sufficiently provided. To analyze important alternative splicing switches, we referred to a study published in *Nucleic Acids Research* that used a newly developed computational method to identify a set of isoform pairs that could distinguish tumor cells from normal cells. (Sebestyen et al., 2015). According to the article, "Relative abundance per sample (or PSI) was calculated by normalizing the TPM to the sum of the TPMs for all transcripts of a gene." Referring to this article, it is regrettable that the terms relative abundance (or frequency) of TPM and PSI may be used interchangeably to confuse readers.

Accordingly, the iso-kTSP algorithm was used to calculate the relative expression of alternative splicing variants containing intron retention and mutually exclusive exons as well as different types of exon skipping events. Consequently, we were able to detect a consistent reversal of relative isoform expression in the ESRP1-high and low groups, and observed alternative splicing alterations more complex than the local patterns of splicing alterations expressed by PSI.

Nevertheless, as suggested by Reviewer #1, the graphs in **Fig. 1i** and **2f** of the previous figures were replaced with the ones showing the relationship between expression of ESRP1 and the PSI of LRRFIP2 exon 7 (**Fig. C**). These results are included in **Fig.1h** and **Supplementary Fig. 3h**.

Fig. C. LRRFIP2 exon 7 skipping event is significantly negatively correlated to ESRP1 expression. **a** The PSI value of LRRFIP2 exon 7 in 18 gastric cancer cell lines ($P=0.0002$)
b Pearson's correlation analysis between the PSI value of LRRFIP2 exon 7 and ESRP1 expression (TPM) in 18 patient tissue samples ($R= -0.7724$, $P=0.0002$)

Reference

Sebestyen, E., Zawisza, M. & Eyras, E. Detection of recurrent alternative splicing switches in tumor samples reveals novel signatures of cancer. *Nucleic Acids Res* 43, 1345-56 (2015).

Reviewer #3 (Remarks to the Author):

The revised version is much improved compared to the original. The authors have addressed the concerns from the reviews well. The only minor concern that remains to me is that the LRRFIP2 knockout, KO #1 and #2 in the figures are independent clones rather than clones from independent sequences, as clarifying in the rebuttal letter. This information should be stated clearly in the manuscript not just in the rebuttal. Since no independent sequence of gRNA has been used, to rule out off target effects from the gRNAs it shall be suggested to perform overexpression of LRRFIP2 v3 in at least one of the KO clones to rescue the effect from the KO.

Response:

As Reviewer#3 pointed out, editing multiple sites with a CRISPR system minimizes the potential for off-target effects. However, we had to knock out a specific exon rather than the entire gene, so there was a limitation to targeting multiple sites. We performed single cell selection in the expectation that different clones would have different genome editing sites. However, the various clones we obtained showed exactly the same sequence after single cell selection. Therefore, we overexpressed LRRFIP2 variant 3 in the knockout clone used in our previous experiments and investigated the rescue effect of the variant as kindly suggested by the reviewers.

Consistent with our previous results, the invasiveness and migratory potential were restored by exogenous introduction of variant 3 (**Fig. Aa**). Furthermore, overexpression of LRRFIP2 variant 3 in exon 7 knockout cell line rescued the expression of SERPINE1 and methylation of H3R17. did (**Fig. Aa and c**). Moreover, the reduced promoter activity of *SERPINE1* was upregulated by rescued expression of LRRFIP2 variant 3 (**Fig. Ad**). Taken together, our results show that metastatic potential of MKN1 gastric cancer cells, reduced by exon 7 deletion, was restored by LRRFIP2 variant 3 overexpression, eliminating the possibility that the phenomenon observed in exon 7 knockout cell lines was caused by off-target editing by CRISPR/Cas9. As suggested by Reviewer#3, we have included this information in manuscript **p.9, line #169-173** and **Supplementary Figs. 11 and 13**.

Fig. A. Overexpression of LRRFIP2 variant in exon 7 KO cell line rescues invasiveness and migration of MKN1 gastric cancer cells. **a** Transwell migration assay and Matrigel invasion assay of the control, exon 7-knockout, and variant 3-rescued cell lines (left) and bar graphs showing number of invaded and migrated cells (right), respectively, following staining with crystal violet. **b** Immunoblot analysis of histone H3R17 methylation and LRRFIP2. **c** RT-PCR analysis of *LRRFIP2* and *SERPINE1* in MKN1 control and exon 7 knockout cells. **d** Luciferase activity of *SERPINE1* promoter following overexpression of LRRFIP2 V3.

REVIEWERS' COMMENTS

Reviewer #1 (Remarks to the Author):

In this revised manuscript, the authors addressed the major concerns raised by the reviewer. Please see below comments before acceptance:

1. The authors analyzed the relationship between ESRP1 expression of alternative splicing. Which tumors were considered ESRP1-high and ESRP1-low shown in Fig. S3?
2. The authors need to carefully evaluate the KEGG and GO pathway analysis in Fig. 1d,1e and Fig. S3d, e. (1) The authors wrote in Line 116 "the upregulated genes in ESRP1-low cell lines were highly enriched for proteoglycans in cancer, ..." However, "Proteoglycans in cancer" was shown in ESRP1-high associated pathway in Fig. 1e. (2) "Regulation of signal transduction" described in the text for Fig 1e was not shown in Fig 1e. (3) Hippo signaling pathway was shown enriched in ESRP1-low cell line data (Fig. 1d) but was associated with ESRP1-high tumors (Fig. S3e). (4) "Regulation of cell division" was associated with ESRP1-high cell lines (Fig. 1e), yet "Positive regulation of cell proliferation" was shown in ESRP1-low tumors (Fig. S3d). These are contradictory findings and should not be used as a statement in text line 126: "Despite of the heterogenic characteristic of gastric cancer tissues, a number of the terms and pathways show consistency with the findings in the cell lines, such as cell junctions, proliferation and motility, supporting the notion that ESRP1 has a critical function in metastasis and tumorigenesis by suppressing tumor motility and invasiveness in gastric cancer cells."- 3. Fig. 1j: Longer exposure of the image needs to be shown to visualize the bands. LRRFIP2 isoforms were not seen in agarose gel image in several cell lines but was quantified in the upper panel of Fig. 1j.
- 4. Fig. S4a,b. Since the sum of isoform A and isoform B equals 1, plotting PSI values vs ESRP1 will make the graphs easier to visualize. PSI, which is percent spliced in, is the relative expression of isoform A in panel a, and isoform B in panel b.
- 5. Fig. S6 can also be plotted as PSI vs ESRP1 expression. In current plot, R value measures the correlation between v2 and v3, regardless of ESRP1 levels. Since the sum of v2 and v3 equals 1, it is expected that the R value is ~ -1 . However, the statistical correlation between PSI and ESRP1 levels should help elucidate the relationship between ESRP1 and LRRFIP exon skipping
- 6. Line 151: "These observations support the notion that isoform switching event of LRRFIP2 may be critical for determination of the epithelial phenotype regulated by ESRP1 not only in gastric cancer cells but also in other types of cancers." This is an overstatement and should be edited. First, the data does not support any causality, and thus "for determination of the epithelial phenotype" should be eliminated. Second, data in Fig. S7 is not strong, expanding the conclusion to other cancer types is not appropriate and not necessary.
- 7. Please indicate the promoter region of SERPINE1 that was used for PCR analysis in ChIP assays?

Minor comments:

1. The legend for Fig. 5h is missing.
2. The authors should check spelling and grammar errors throughout the manuscript.

Reviewer #3 (Remarks to the Author):

My comments have been addressed.

ESRP1-regulated isoform switching of LRRFIP2 determines metastasis of gastric cancer

#NCOMMS-21-22414B

Response to Reviewers

We greatly appreciate the constructive comments from the reviewers and the invitation from the editor to submit a revised version. We have thoroughly revised the manuscript following the reviewer's suggestions. Please see below our point-to-point responses in non-italic text following reviewer comments in italic text.

REVIEWERS' COMMENTS

Reviewer #1 (Remarks to the Author):

In this revised manuscript, the authors addressed the major concerns raised by the reviewer. Please see below comments before acceptance:

1. The authors analyzed the relationship between ESRP1 expression of alternative splicing. Which tumors were considered ESRP1-high and ESRP1-low shown in Fig. S3?

Response: 136T, 87T, 236T, 211T, 80T, 135T, and 134 T were considered ESRP1-low and 130T, 103T, 95T, 195T, 849T, 43T, 917T, 859T, 119T, 889T, and 882T were considered ESRP1-high. This information was written in Materials and Methods, but we also added it in the manuscript line78-82, p6.

2. The authors need to carefully evaluate the KEGG and GO pathway analysis in Fig. 1d, 1e and Fig. S3d, e. (1) The authors wrote in Line 116 "the upregulated genes in ESRP1-low cell lines were highly enriched for proteoglycans in cancer, ..." However, "Proteoglycans in cancer" was shown in ESRP1-high associated pathway in Fig. 1e. (2) "Regulation of signal transduction" described in the text for Fig 1e was not shown in Fig 1e. (3) Hippo signaling pathway was shown enriched in ESRP1-low cell line data (Fig. 1d) but was associated with ESRP1-high tumors (Fig. S3e). (4) "Regulation of cell division" was associated with ESRP1-high cell lines (Fig. 1e), yet "Positive regulation of cell proliferation" was shown in ESRP1-low tumors (Fig. S3d). These are contradictory findings and should not be used as a statement in text line 126: "Despite of the heterogenic characteristic of gastric cancer tissues, a number of the terms and pathways show consistency with the findings in the cell lines, such as cell junctions, proliferation and

motility, supporting the notion that ESRP1 has a critical function in metastasis and tumorigenesis by suppressing tumor motility and invasiveness in gastric cancer cells."

Response: We corrected the KEGG and GO pathway analysis carefully. (1&2) We made correction. (3) Hippo signaling pathway seems to be associated with ESRP1 expression. Thus, it was observed in ESRP1-low and high cell line data and ESRP1-high tumor data. (4) We excluded "Regulation of cell division" and "Positive regulation of cell proliferation" from the figures, since these terms which are not critical for our hypothesis might confuse readers.

We also adjusted the conclusion of KEGG and GO analysis: "Although there is some discrepancy in the terms and pathways between cell lines and tumor tissues due to the heterogenic characteristic of gastric cancer tissues, the terms and pathways regarding cell junctions and motility show consistency supporting the notion that ESRP1 has a critical function in metastasis by suppressing tumor motility and invasiveness in gastric cancer cells."

3. Fig. 1j: Longer exposure of the image needs to be shown to visualize the bands. LRRFIP2 isoforms were not seen in agarose gel image in several cell lines but was quantified in the upper panel of Fig. 1j.

Response: We have replaced the image with longer exposure time, as suggested.

4. Fig. S4a, b. Since the sum of isoform A and isoform B equals 1, plotting PSI values vs ESRP1 will make the graphs easier to visualize. PSI, which is percent spliced in, is the relative expression of isoform A in panel a, and isoform B in panel b.

Response: As suggested by the reviewer, we replaced the graphs with the ones plotting PSI values vs ESRP1 in Fig. S4a and b (**Figure A**).

Figure A. Exon skipping events of CCDC50 and BICD2 are correlated with ESRP1 expression. **a** Pearson's correlation analysis between the PSI of CCDC50 (NM_178335)

exon 6 in the association of ESRP1(TPM) from Supplementary Fig. 2b and c ($R=0.6562$, $P=0.0031$). b Pearson's correlation analysis between the PSI of BICD2 (NM_015250) exon 8 in the association of ESRP1(TPM) from Supplementary Fig. 2b and c ($R=-0.6327$, $P=0.0048$).

5. Fig. S6 can also be plotted as PSI vs ESRP1 expression. In current plot, R value measures the correlation between v2 and v3, regardless of ESRP1 levels. Since the sum of v2 and v3 equals 1, it is expected that the R value is ~ -1 . However, the statistical correlation between PSI and ESRP1 levels should help elucidate the relationship between ESRP1 and LRRFIP exon skipping

Response: We calculated the relative TPM of the variants among four variants (Fig. S4a) in Fig. S6, rather than PSI. The data representing the relation between ESRP1 and PSI of LRRFIP2 variant 3 exon 7 is in Fig. 1h, we excluded Fig. S6 from the supplementary figures.

6. Line 151: "These observations support the notion that isoform switching event of LRRFIP2 may be critical for determination of the epithelial phenotype regulated by ESRP1 not only in gastric cancer cells but also in other types of cancers." This is an overstatement and should be edited. First, the data does not support any causality, and thus "for determination of the epithelial phenotype" should be eliminated. Second, data in Fig. S7 is not strong, expanding the conclusion to other cancer types is not appropriate and not necessary.

Response: We agree that the data is not strong and not necessary to support our overall hypothesis. Thus, we excluded Fig. S7 from the supplementary figures.

7. Please indicate the promoter region of SERPINE1 that was used for PCR analysis in ChIP assays?

Response: We have included the promoter region used for PCR analysis in the figure legend.

Minor comments:

1. The legend for Fig. 5h is missing.

Response: We appreciate your comment on the missing figure legend. We have added the

legend for Fig. 5h.

2. *The authors should check spelling and grammar errors throughout the manuscript.*

Response: We checked spelling and grammar errors.